# Tunnelling to holographic traversable wormholes

**Suzanne Bintanja[1,2⋆], Ben Freivogel[1,3†] and Andrew Rolph[1‡]**

**1** Institute for Theoretical Physics, University of Amsterdam,
Science Park 904, 1090 GL Amsterdam, The Netherlands
**2** Kavli Institute for Theoretical Physics, University of California,
Santa Barbara, CA 93106, USA
**3** GRAPPA, University of Amsterdam, Science Park 904,
1090 GL Amsterdam, The Netherlands

⋆ s.bintanja@uva.nl , † b.w.freivogel@uva.nl , ‡ andrew.d.rolph@gmail.com

## Abstract

We study nonperturbative effects of quantum gravity in a system consisting of a coupled pair of holographic CFTs. The $AdS_4/CFT_3$ system has three possible ground states: two copies of empty AdS, a pair of extremal AdS black holes, and an eternal AdS traversable wormhole. We give a recipe for calculating transition rates via gravitational instantons and test it by calculating the emission rate of radiation shells from a black hole. We calculate the nucleation rate of a traversable wormhole between a pair of AdS-RN black holes in the canonical and microcanonical ensembles. Our results give predictions of nonpertubative quantum gravity that can be tested in a holographic simulation.

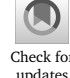

# 1 Introduction

Without a Planck energy collider, it is unclear if or when we will directly observe quantum gravity in the real world. However, we can make use of gauge/gravity duality to perform quantum gravity experiments on quantum computers in the near future. By simulating quantum theories that are dual to gravity in asymptotically AdS spacetime, we can test our gravity predictions.

In this paper, we consider a system where nonperturbative quantum gravity effects can be studied. The system is holographic with two adjustable parameters. The boundary dual theory is a coupled pair of CFTs with a coupling parameter $h$ and a chemical potential $\mu$ that controls the charge of the system. The bulk dual theory is Einstein-Maxwell with a negative cosmological constant and a charged fermion. By varying the coupling and the chemical potential, there are three possible ground states: a traversable wormhole, a pair of extremal AdS Reissner Nordström (RN) black holes, and empty AdS.

If we prepare the system in the state where the ground state is a pair of extremal black holes and then dial couplings until the ground state is a traversable wormhole, the bulk geometry will presumably transition dynamically. With mild assumptions, dynamical topology change in semiclassical gravity is forbidden [1–3]. To evolve from a pair of disconnected extremal black holes to a traversable wormhole requires a change in topology, so the dominant decay channel will be quantum mechanical; tunnelling via gravitational instanton. We want to find the tunnelling rate and the trajectory of the domain wall after the tunnelling event. By using instanton techniques, we calculate transition rates between the black hole and traversable wormhole geometries in the microcanonical and canonical ensembles. In other words, we compute the non-perturbative, yet dominant decay rate between semi-classical solutions with different topologies. Hence, if this model can be simulated, then these non-perturbative results can in principle be tested experimentally.

Quantum effects are fundamental to the existence of traversable wormholes, as well as for tunnelling between geometries. The traversable wormhole needs to be supported by negative null energy, which in our holographic model comes from a quantum Casimir effect; traversability requires a violation of the averaged null energy condition [4,5]. The existence of traversable wormholes is less speculative than transitions through topology-changing fluctuations because the wormhole solutions do not require control over nonperturbative gravitational effects, but only that we can couple gravity to quantum fields within the framework of semi-classical gravity.

On the other hand, the regime of validity of the semiclassical approximation is not fully understood, and breaks down in non-trivial ways, for example, for long wormholes, and it would be interesting to test predictions outside of the semiclassical regime with measurements of the boundary dual.

The tunnelling to and real-time production rate of traversable wormholes was previously calculated in other settings in [6–8]. Compared to our work, there are similarities in motivation, but also fundamental differences between the models within which the calculations are done. The papers [7,8] study the problem in a lower dimensional SYK/JT gravity model, and the basic mechanism of wormhole production in [6], the breaking of cosmic strings, is different from ours.

**Summary of results**

In section 2, we review the traversable wormhole model of [9], give a new derivation of the wormhole mass, and calculate the coupling at which the semiclassical approximation breaks down. Our traversable wormhole model was introduced in [9]. The boundary Hamiltonian couples a pair of CFTs through the operators $\Psi_{\pm}^{L,R}$ dual to the bulk fermion and has a chemical potential term that energetically favours charge differences:

$$H := H_L + H_R - \frac{ih}{\ell} \int d\Omega_2 \left( \overline{\Psi}_-^R \Psi_+^L + \overline{\Psi}_+^L \Psi_-^R \right) + \mu(Q_L - Q_R). \tag{1}$$

The mechanism that sources the negative energy density in the bulk was inspired by a construction in asymptotically flat spacetime by Maldacena, Milekhin, and Popov [10]. In our

opinion, our setup offers several advantages. Firstly, our wormhole is eternal and asymptotically AdS$_4$, and embedding the wormhole in AdS makes tests using AdS/CFT possible. This is a feature that our model shares with the models of Gao, Jafferis, and Wall and the eternal asymptotically AdS$_2$ model of Maldacena and Qi [11,12]. Secondly, our traversable wormhole is the ground state of a simple Hamiltonian, which lends itself to preparation in the lab.

We are also interested in the phase structure of our model. With a view to more precisely mapping out the phase boundary between our wormholes and empty AdS, we extend the results of [9] to study wormholes with small charges, whose size is much smaller than the AdS radius. We determine $\Delta M$, the mass deficit of the wormhole solution with respect to a pair of extremal AdS-RN black holes with the same $U(1)$ charge $Q$, to be

$$
\Delta M \sim
\begin{cases}
-\dfrac{\lambda(h)Q}{\ell}, & Q^2 \ll \ell m_P \lambda, \\[2ex]
-\dfrac{\lambda(h)^2 m_p}{Q}, & Q^2 \gg \ell m_P \lambda, \quad \text{and} \quad \bar{r} \ll \ell,
\end{cases}
\tag{2}
$$

where $\lambda(h)$, given in (18), is a function of the non-local coupling, $\ell$ is the AdS length, and $\bar{r}$ is the horizon radius of the extremal AdS-RN black hole. Additionally, in the large charge, large black hole limit we recover the typical $Q^{3/2}$ scaling. More precise equations for the wormhole mass are presented in section 2.1.

In section 3, we discuss the phases of our model in the canonical and microcanonical ensembles and discuss fragmentation. We determine in which regimes of parameters the wormhole is the ground state, see figure 1.

We also compute the temperature at which the traversable wormhole begins to dominate in the regime of parameters where it is the ground state. Although the traversable wormhole is the ground state, the black hole has a much larger entropy, so the wormhole only dominates

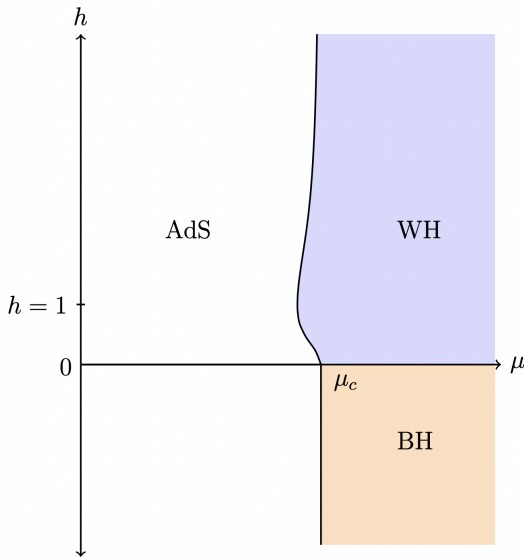

Figure 1: A phase diagram of the ground states of our model for different values of boundary coupling $h$ and chemical potential $\mu$, and the corresponding bulk geometries: a pair of empty AdS spacetimes, an asymptotically AdS traversable wormhole, and a pair of extremal AdS-RN black holes. The identification of boundary states with particular bulk geometries through the semiclassical approximation breaks down near the phase boundaries in a way we make precise later.

the canonical ensemble below the transition temperature $T_c \sim S_{BH}^{-1}$. We find that this is below the temperature at which the semi-classical description of the black hole breaks down. The wormhole itself is well described semi-classically, but attempting to cool the black hole to a low enough temperature that it wants to transition to a wormhole takes us out of the semi-classical regime.

Despite this issue, in the following sections, we give a first estimate of the transition rate from the black hole to the wormhole. Typically it is convenient to compute tunnelling in the thin wall approximation. However, it is unclear what sort of domain wall should connect the two geometries. We propose a simple domain wall with a 'radiation' equation of state. This type of domain wall is suitable for describing transitions between two states that do not differ by any conserved charges, and are also not separated by potential barriers.

In section 4, we give the recipe to compute transition rates. We first test our techniques by calculating transition rates between charged black holes. Similar techniques have been used in computations of Hawking radiation as tunnelling [13–16]. However, our actual instantons differ from these references due to our use of a thin wall with radiation equation of state. It is not obvious to us how to understand the results in the literature in the framework of the thin wall approximation. We are not questioning these results, but simply offering a different framework for computing transition rates. Our results agree with where we have compared them. Our instantons are static Euclidean solutions that interpolate between black holes of different masses.[1] For Schwarzschild black holes in asymptotically flat spacetime, the domain wall sits at $r = 3GM$. From the Lorentzian point of view, radiation first tunnels to some radius outside the black hole, and then classically moves out to infinity.

In section 5, we calculate the nucleation rates of traversable wormholes in both the canonical and microcanonical ensembles. We describe the instanton that is relevant for tunnelling from black holes to wormholes and compute the tunnelling rate. Our fixed energy result is perhaps unsurprising,

$$\Gamma \sim e^{-2S_{BH}} \, . \tag{3}$$

The factor of 2 is due to the fact that the black hole ground state consists of two black holes. We also determine the rate for a pair of finite temperature AdS-RN black holes to nucleate a traversable wormhole connecting the geometries, by calculating the difference in on-shell actions,

$$\Gamma(\beta) \sim e^{-2S_{BH}+2\beta(M_{BH}-M_o)} \, . \tag{4}$$

$M_{BH}$ and $S_{BH}$ are the mass and entropy of a single AdS-RN black hole at inverse temperature $\beta$, and $M_o$ is the asymptotic mass of the geometry after the nucleation of the traversable wormhole, which is not fixed in the canonical ensemble. The transition rate exponent is proportional to the change in free energy.

Lastly, in section 6, we summarise the main conclusions and discuss some of the many remaining open questions and future directions.

## 2 AdS traversable wormholes from coupled CFTs

In this section we review the construction of the traversable wormhole solution found in [9], and extend some of the results. We present an alternative derivation of the change in energy due to the non-local coupling, and we discuss the breakdown of the semiclassical approximation we use when the wormhole throat becomes long.

---

[1]In many textbook examples, such as in [17], instantons are localised in Euclidean time, as their name suggests. Nonetheless, static instantons, such as the Hawking-Moss instanton [18], exist and are important [19,20].

## 2.1 Traversable wormhole solution

The ingredients of our traversable wormhole solution are the following. We consider the Einstein-Maxwell theory with a negative cosmological constant, together with a $U(1)$-gauge field and a massless Dirac fermion coupled to the gauge field. A well-known solution to this theory is the extremal magnetically charged Reissner-Nordström (RN) black hole in AdS. This solution can be cast as a wormhole, albeit one that is not traversable, since spacetime has a singularity. Nevertheless, the extremal RN-AdS black hole forms an important stepping stone in the construction of the traversable wormhole, since its near-horizon geometry is $\text{AdS}_2 \times S^2$. Because the sphere has a constant radius, only a small amount of negative energy is needed to defocus null geodesics, causing the sphere to re-expand. This renders the wormhole traversable.

The bulk action of the theory is given by

$$S_{\text{bulk}} = \int d^4x \sqrt{g} \left( \frac{1}{16\pi G} \left( R - \frac{6}{\ell^2} \right) - \frac{1}{4g^2} F^2 + i \overline{\Psi} \slashed{D} \Psi \right), \tag{5}$$

where we consider the gauge coupling $g$ to be small, which implies that loop corrections are negligible. We will work in a spherically symmetric static Ansatz with a nonzero magnetic field for the geometry that is parametrized by the integer $q$:

$$ds^2 = e^{2\sigma(x)}(-dt^2 + dx^2) + R^2(x) \, d\Omega_2^2, \qquad A = \frac{q}{2} \cos(\theta) d\phi. \tag{6}$$

The radial coordinate $x$ should be thought of as compact, and setting the range of it can be seen as a gauge choice. We set $x \in [-\frac{\Delta x}{2}, \frac{\Delta x}{2}]$. Note that this geometry has no horizons or singularities. We will be interested in solutions that have two asymptotically AdS regions: one in each of the regions approaching $x = \pm \frac{\Delta x}{2}$. Therefore, a consistent solution to the equations of motion constitutes an eternal traversable wormhole in AdS.

### 2.1.1 Bulk fermion decomposition and equation of motion

We will use the following Ansatz for the Dirac fermion that allows us to decompose solutions to the equations of motions on the sphere[2,3]

$$\Psi(t, x, \theta, \phi) = \frac{e^{-\frac{\sigma(x)}{2}}}{R(x)} \sum_m \psi_m(t, x) \otimes \eta_m(\theta, \phi), \tag{7}$$

with $\psi$ and $\eta$ bi-spinors. We will denote the components of $\psi$ with $\psi_{\pm}$, and choose the eigenvectors of $\sigma_z$ as basis for $\eta$, i.e. $\eta_{\pm}\sigma_z = \pm\eta_{\pm}$. In this Ansatz, the Dirac equation reduces to

$$\frac{e^{-\frac{3}{2}\sigma(x)}}{R(x)} \left( i\sigma_x \partial_t + \sigma_y \partial_x \right) \psi \otimes \eta = -\lambda,$$

$$\frac{e^{-\frac{\sigma}{2}}}{R^2} \sigma_z \psi \otimes \left( \sigma_y \frac{\partial_\phi - iA_\phi}{\sin(\theta)} + \sigma_x \left( \partial_\theta + \frac{1}{2}\cot(\theta) \right) \right) \eta = \lambda, \tag{8}$$

where $\sigma_i$ denotes the $i$-th Pauli matrix, and different values of $\lambda$ illustrate the splitting into Landau levels. For the lowest Landau level (which we will mostly be concerned with), given by $\lambda = 0$, the solutions are

$$\psi_{\pm} = \sum_k \frac{\alpha_k^{\pm}}{\sqrt{\pi \Delta x}} e^{i\omega_k x_{\mp}}, \quad \eta_{\pm}^m = \left( \sin\frac{\theta}{2} \right)^{j_{\pm} \pm m} \left( \cos\frac{\theta}{2} \right)^{j_{\pm} \mp m} e^{im\phi},$$

$$\text{where} \quad j_{\pm} = \frac{1}{2}(-1 \mp q). \tag{9}$$

---

[2]We will suppress indices whenever possible.

[3]See appendix A for our conventions regarding the vielbein and gamma matrices.

Here we have used lightcone coordinates $x_\pm := t \pm x$ and we have introduced shorthand notation $\overline{\Delta x} := \Delta x/\pi$. If we take the integer $q$ to be greater than zero, the solution on the sphere is given by

$$\eta_+ = 0, \quad \text{and} \quad \eta_- = \sum_m \mathcal{C}_m^j \eta_-^m, \tag{10}$$

where $j = j_-$, and $m \in \{-j_-, -j_- + 1 \cdots, j_- - 1, j_-\}$. We fix the normalization $\mathcal{C}_m^j$ such that

$$\int d^2\Omega \, \overline{\eta}^m \eta^n = \delta_{m,n}. \tag{11}$$

### 2.1.2 Boundary couplings and boundary conditions

In order to have a well-defined variational principle, we add boundary terms to the action that impose boundary conditions on exactly half the degrees of freedom. Furthermore, we add a non-local boundary term that couples the two boundaries and modifies the classical boundary conditions; this will provide an NEC-violating stress tensor, rendering the wormhole traversable. The boundary action is simplest when we project the 4D Dirac spinor onto the eigenspace of the gamma matrix in the holographic direction

$$\Psi_\pm := \mathcal{P}_\pm \Psi_\pm, \quad \text{where} \quad \mathcal{P}_\pm := \frac{1}{2}\left(1 \pm \gamma^2\right). \tag{12}$$

The boundary action we pick is given by

$$S^\partial := S^\partial_{\text{classical}} + S^\partial_{\text{non-local}}, \tag{13}$$

with

$$S^\partial_{\text{classical}} = i \int_\partial d^3x \sqrt{\gamma} \left(\overline{\Psi}_+^L \Psi_-^L - \overline{\Psi}_-^R \Psi_+^R\right), \tag{14}$$

$$S^\partial_{\text{non-local}} = ih \int_\partial d^3x \sqrt{\gamma} \left(\overline{\Psi}_-^R \Psi_+^L + \overline{\Psi}_+^L \Psi_-^R\right). \tag{15}$$

Here $h \in \mathbb{R}$ is the coupling constant for the non-local interaction. With this choice of boundary action, the boundary conditions are given by

$$\Psi_+^R + h\Psi_+^L = 0, \quad \text{and} \quad \Psi_-^L + h\Psi_-^R = 0. \tag{16}$$

A simple computation shows that under these boundary conditions, no energy or charge leaks out at the boundary, as is required in any consistent solution.

### 2.1.3 Bulk fermion solution

Using the above boundary conditions, we can solve the equations of motion in the time and radial directions, leading to the following frequencies and modes

$$\alpha_k^+ = (-1)^{k+1}\alpha_k^-, \quad \text{and} \quad \overline{\Delta x}\,\omega_k = \frac{2k+1}{2} + (-1)^k \frac{2\lambda(h)}{\pi}, \tag{17}$$

with $\lambda(h)$ a function of $h$ defined as

$$\lambda(h) := \frac{1}{2}\arctan\left(\frac{2h}{|1-h^2|}\right). \tag{18}$$

Interestingly, the solution is invariant under $h \mapsto 1/h$. Therefore, the solution exhibits a form of S-duality. By using the above solutions, imposing canonical commutation relations for the fermionic fields, and defining a vacuum as

$$\alpha_k |0\rangle = 0, \quad \forall k \in \mathbb{Z}_{<0}, \quad \text{and} \quad \alpha_k^\dagger |0\rangle = 0, \quad \forall k \in \mathbb{Z}_{\geq 0}, \tag{19}$$

we can evaluate the fermionic two-point functions with the following result:[4]

$$\langle \psi_+^\dagger(x_-)\psi_+(x'_-)\rangle = \frac{1}{\pi \Delta x} \frac{e^{-i\frac{x'_- - x_-}{\Delta x}\left(\frac{1}{2} + \frac{2i\lambda(h)}{\pi}\right)} + e^{i\frac{x'_- - x_-}{\Delta x}\left(\frac{1}{2} + \frac{2i\lambda(h)}{\pi}\right)}}{1 - e^{2i\frac{x'_- - x_-}{\Delta x}}},$$

$$\langle \psi_-^\dagger(x_+)\psi_-(x'_+)\rangle = \frac{1}{\Delta x \pi} \frac{e^{-i\frac{(x'_+ - x_+)}{\Delta x}\left(\frac{1}{2} + \frac{2i\lambda(h)}{\pi}\right)} + e^{i\frac{x'_+ - x_+}{\Delta x}\left(\frac{1}{2} + \frac{2i\lambda(h)}{\pi}\right)}}{1 - e^{2i\frac{x'_+ - x_+}{\Delta x}}}, \tag{20}$$

$$\langle \psi_+^\dagger(x_-)\psi_-(x'_+)\rangle = \frac{1}{\Delta x \pi} \frac{-e^{-i\frac{x'_+ - x_-}{\Delta x}\left(\frac{1}{2} + \frac{2i\lambda(h)}{\pi}\right)} + e^{i\frac{x'_+ - x_-}{\Delta x}\left(\frac{1}{2} + \frac{2i\lambda(h)}{\pi}\right)}}{1 - e^{2i\frac{x'_+ - x_-}{\Delta x}}}.$$

We can use these two-point functions to obtain the change in the stress tensor due to the non-local coupling through point-splitting. Because our metric Ansatz (6) is static and spherically symmetric, the only off-diagonal component of the stress tensor that can be nonzero is $T_{12}$. Furthermore, from the tracelessness of the 4d stress tensor, and the tracelessness of the 2d stress tensor obtained after dimensional reduction on the sphere, the only diagonal components of the stress tensor that can be nonzero are $T_{11} = T_{22}$. The point-splitting procedure results in

$$\left\langle T_{\mu\nu}^h \right\rangle - \left\langle T_{\mu\nu}^{h=0} \right\rangle = -\frac{1}{2\pi^3} \frac{q\lambda(h)}{\Delta x^2 R^2(x)} \text{diag}(1,1,0,0). \tag{21}$$

One thing to notice is that the charge $q$ of the black hole appears as a linear factor. This factor $q$ comes from the fact that when we dimensionally reduce, the four-dimensional fermionic field leads to $q$ effectively massless 2d fermions, each of which contributes to the stress tensor. We can use this fact to enhance the negative energy while at the same time only having to non-locally couple one 4d field.

### 2.1.4 Wormhole geometry solution

We can now finally solve for the wormhole geometry sourced by the above stress tensor (21) through the semiclassical Einstein equations. Note that the geometry has to be consistent with our Ansatz (6). It turns out that we can solve the linearized Einstein equations analytically when the parameter $\zeta := \frac{4Gq\lambda(h)}{\pi^2 \Delta x^2 \bar{r}^2}$ is small.[5] Here

$$\bar{r}^2 = \frac{\ell^2}{6}\left(\sqrt{1 + 12\frac{r_e^2}{\ell^2}} - 1\right), \tag{22}$$

is the horizon radius of the extremal RN-AdS black hole with magnetic charge $r_e = \frac{\sqrt{\pi G}q}{g}$. Deep inside the wormhole throat, the metric takes the form of $\text{AdS}_2 \times S^2$ with a small deformation:

$$ds^2 = \frac{\bar{r}^2}{\mathcal{C}(\bar{r})}\left(-\left(1 + \rho^2 + \gamma(\rho)\right)dt^2 + \frac{d\rho^2}{1 + \rho^2 + \gamma(\rho)}\right) + \bar{r}^2\left(1 + \psi(\rho)\right)d\Omega_2^2, \tag{23}$$

---

[4]Compared to [9] here the correlators are computed at arbitrary separation.

[5]We can solve the full equations numerically. In the linearized limit, the numerical and analytical solutions agree.

where

$$\mathcal{C}(\bar{r}) := \frac{6\bar{r}^2}{\ell^2} + 1 \, . \tag{24}$$

The functions $\psi$ and $\gamma$ denote the deformation away from AdS$_2 \times S^2$, and are given by

$$\psi(\rho) = \zeta(1 + \rho \arctan(\rho)) \, , \tag{25}$$

and

$$\gamma(\rho) = -\frac{\zeta\left(1 + 4\frac{\bar{r}^2}{\ell^2}\right)}{\mathcal{C}(\bar{r})} \left(\rho^2 + \rho(3 + \rho^2)\arctan(\rho) - \log\left(1 + \rho^2\right)\right) \, . \tag{26}$$

We have chosen coordinates so that the $t$ coordinate matches between the metrics (23) and (6). Note that indeed these functions are small when $\zeta$ is small. This geometry deep inside the wormhole throat can be matched onto the near-horizon limit of a super-extremal RN-AdS black hole

$$ds^2 = -f(r)d\tau^2 + \frac{dr^2}{f(r)} + r^2 d\Omega_2^2 \, , \tag{27}$$

with

$$f(r) = \frac{r^2}{\ell^2} + 1 - \frac{2GM}{r} + \frac{r_e^2}{r^2} \, . \tag{28}$$

Here $M$, and $r_e$ denote the asymptotic charges. The matching relates the coordinates $\tau$ and $r$ on the outside to the coordinates $t$ and $\rho$ used in the throat

$$t = \frac{\mathcal{C}(\bar{r})\tau}{L} \, , \quad \text{and} \quad \rho = \frac{4(r - \bar{r})}{\pi\bar{r}\zeta} \, , \tag{29}$$

where $L$ is an integration constant that denotes up to what $\rho$ we can trust the throat geometry, *i.e.* the $\rho$ coordinate has a cutoff at $\rho \sim \frac{L}{\bar{r}}$. In other words: $L$ measures the length of the wormhole, and is given by

$$L = \frac{4\bar{r}}{\pi\zeta} \, , \tag{30}$$

which implies

$$\rho = \frac{L(r - \bar{r})}{\bar{r}^2} \, . \tag{31}$$

The effect of the non-local coupling can be seen in the asymptotic mass, which is equal to the mass of the extremal RN-AdS black hole $M_{\text{ext}}$ together with a negative contribution $\Delta M$, so that

$$M_{\text{WH}} = M_{\text{ext}} + \Delta M \, , \quad \text{with} \quad M_{\text{ext}} = \frac{\bar{r}}{G} + \frac{2\bar{r}^3}{G\ell^2} \, . \tag{32}$$

We will find the value of $\Delta M$ after we fix the gauge by computing $\overline{\Delta x}$ in section 2.1.6. To simplify the upcoming discussions, we first describe the geometry and split it into three convenient regions.

### 2.1.5 Far, mouth and throat regions of the wormhole geometry

Before we discuss the different regions in the geometry we must introduce the geometry in the near horizon limit of a super-extremal RN black hole. The near horizon metric can be found by analyzing the zeros of the fourth-order polynomial $r^2 f(r)$, with $f$ given in (28). This polynomial has four zeroes; in the case of the super-extremal black hole, they are two pairs of complex conjugates. We parametrize one pair of zeroes by $r_{1,2} = \hat{r}(1 \pm i\epsilon)$, with $\epsilon \in \mathbb{R}_{>0}$. Here $\epsilon$ measures the 'distance' from extremality, *i.e.* $\epsilon = 0$ corresponds to an extremal black hole. The remaining roots $r_{3,4}$ can then be found by comparing

$$r^2 f(r) = \frac{1}{\ell^2} \prod_{i=1}^{4} (r - r_i), \tag{33}$$

to (28). In the near-extremal limit, which is given by $\epsilon \ll 1$, the remaining parameters $\hat{r}$, and $r_{3,4}$ take the simple form

$$\hat{r} = \bar{r} - \epsilon^2 \frac{\bar{r}}{2\mathcal{C}(\bar{r})} \left( 1 + 2 \frac{\bar{r}^2}{\ell^2} \right) + \mathcal{O}(\epsilon^4),$$
$$(r - r_3)(r - r_4) = \ell^2 + r^2 + 2r\bar{r} + 3\bar{r}^2 - \frac{\ell^2(r + 4\bar{r}) + 2\bar{r}^2(r + 6\bar{r})}{\ell^2 + 6\bar{r}^2} \bar{r}\epsilon^2 + \mathcal{O}(\epsilon^4). \tag{34}$$

With this parametrization of the emblackening factor, we can easily find the near-horizon geometry, by expanding in $\frac{r-\bar{r}}{\bar{r}} \ll 1$

$$f(r) = \mathcal{C}(\bar{r})\epsilon^2 + \mathcal{C}(\bar{r}) \left( \frac{r-\bar{r}}{\bar{r}} \right)^2 - \mathcal{C}(\bar{r}) \left( \frac{r-\bar{r}}{\bar{r}} \right) \epsilon^2 - 2 \left( 1 + 4\frac{\bar{r}^2}{\ell^2} \right) \left( \frac{r-\bar{r}}{\bar{r}} \right)^3 + \cdots, \tag{35}$$

where we have kept terms up to third order in $\epsilon$ and $\frac{r-\bar{r}}{\bar{r}}$ combined.

By comparing the near-extremal, near-horizon geometry to the wormhole throat geometry we can identify $\epsilon$ with the dimensionless inverse wormhole length

$$\epsilon = \frac{\bar{r}}{L}. \tag{36}$$

Therefore, the longer the wormhole, the closer to extremality the asymptotic black hole geometry is. The matching to the throat geometry of the wormhole in (29) is precisely in the near-extremal, near-horizon limit of the super-extremal RN black hole geometry.

Now that we found the black hole geometry that matches the wormhole throat, we can specify the different regions of the wormhole geometry. The geometry can be split into three regions:

**I: Far region** In this region the radial coordinate is $r$ and it lives in the range $r < \infty$ extending all the way down to the mouth region that we describe next. The geometry is given by (27) with emblackening factor (28) and mass $M_{\text{WH}}$.

**II: Mouth region** This is the region where the asymptotic and throat geometries match, *i.e.* it can be described by both (23) and (27) with emblackening factor (35). This region can be specified by $\epsilon \ll \frac{r-\bar{r}}{\bar{r}} \ll 1$, or equivalently $1 \ll \rho \ll \frac{1}{\epsilon} = \frac{L}{\bar{r}}$.

**III: Throat region** This is the region where there is no description in terms of the usual black hole metric (27). Only the wormhole throat geometry given by the deformation of $\text{AdS}_2 \times S^2$ given in (23) describes this region. The region can be specified by $\rho > 0$ up to the throat region.

The wormhole geometry with the regions described above is depicted in figure 2.

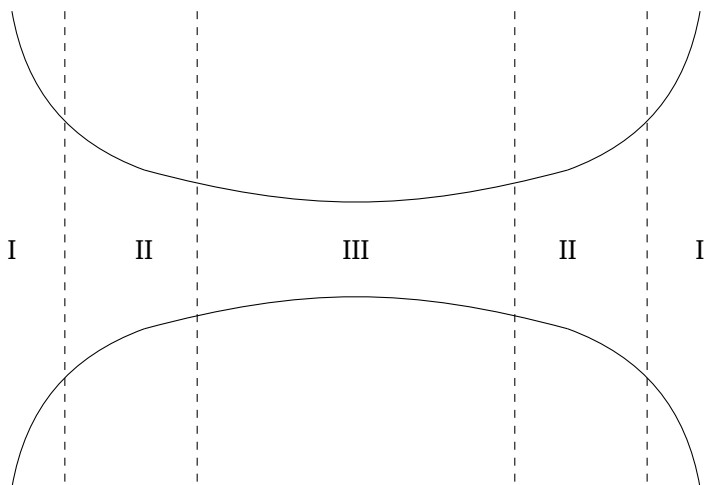

Figure 2: Diagram showing the different regions of the wormhole geometry.

### 2.1.6 Fixing the gauge $\overline{\Delta x}$

Now we need to compute $\overline{\Delta x}$. Comparing the metrics given above, we see that $x$ is related to $\rho$ via

$$\frac{d\rho}{1+\rho^2+\gamma} = dx. \tag{37}$$

The length $\Delta x$ of the throat and mouth region is

$$\Delta x_{throat} = 2\int_0^{\rho_m} \frac{d\rho}{1+\rho^2+\gamma} \approx 2\int_0^\infty \frac{d\rho}{1+\rho^2} = \pi, \tag{38}$$

where we have used that the matching radius $\rho_m$ is large.

Now how about the region outside the throat? In this region, the metric takes the form

$$ds^2 = -f\,d\tau^2 + \frac{dr^2}{f} + r^2 d\Omega_2^2. \tag{39}$$

Compared to the $x$ metric, we must have

$$\frac{dr}{f} = \alpha dx, \tag{40}$$

where $\alpha$ is a constant to be determined. Using this definition,

$$ds^2 = \alpha^2 f(-\alpha^{-2}d\tau^2 + dx^2) + r^2 d\Omega_2^2. \tag{41}$$

To determine $\alpha$, we use the matching formulas (29) together with the relation between $x$ and $\rho$ to obtain

$$dx \approx \frac{d\rho}{\rho^2} \approx \frac{\bar{r}^2 dr}{L(r-\bar{r})^2}, \quad \text{in region II.} \tag{42}$$

On the other hand, since

$$f \approx \mathcal{C}(\bar{r})\left(\frac{r-\bar{r}}{\bar{r}}\right)^2 \approx \mathcal{C}\rho^2\bar{r}^2/L^2, \tag{43}$$

in the matching region II, we require

$$dx = \frac{dr}{\alpha f} \approx \frac{L d\rho}{\alpha \mathcal{C} \rho^2} \,. \tag{44}$$

By comparing to the previous formula for $dx$, this fixes

$$\alpha = \frac{L}{\mathcal{C}} \,. \tag{45}$$

We can now compute the 'length' $\Delta x$ in the far region

$$\Delta x_{far} = 2 \int_{r_m}^{\infty} \frac{dr}{\alpha f} = 2 \frac{\mathcal{C}}{L} \int_{r_m}^{\infty} \frac{dr}{f} \,. \tag{46}$$

There are two main contributions, one near the matching radius $r_m$ and the other at larger $r$. The function $f$ can be simplified in these two regimes, so

$$\Delta x_{far}/2 \approx \frac{1}{L} \int_{r_m}^{r_*} \frac{dr \, \bar{r}^2}{(r - \bar{r})^2} + \frac{\mathcal{C}}{L} \int_{r_*}^{\infty} \frac{dr}{1 + \frac{r^2}{\ell^2}} \,. \tag{47}$$

The matching regime is $\rho_m \sim L/\bar{r}$ which implies

$$\frac{r_m - \bar{r}}{\bar{r}} \sim 1 \,. \tag{48}$$

The small $r$ part of the integral gives order $\frac{\bar{r}}{L}$, which is small and can be neglected. For the large $r$ part, we make a small error by approximating $r_m \approx \bar{r}$, giving

$$\Delta x_{far}/2 \approx \frac{\mathcal{C}}{L} \int_{\bar{r}}^{\infty} \frac{dr}{1 + \frac{r^2}{\ell^2}} \,. \tag{49}$$

This can clearly be computed exactly, but it is more convenient to compute it approximately in two regimes:

$$\Delta x_{far}/2 \approx \begin{cases} \dfrac{\mathcal{C}\ell^2}{\bar{r}L}, & \bar{r} \gg \ell \,, \\[2ex] \dfrac{\pi \mathcal{C}\ell}{2L}, & \bar{r} \ll \ell \,. \end{cases} \tag{50}$$

In the large $\bar{r}$ regime, using $L = 4\bar{r}/(\pi\zeta)$ and $\mathcal{C} \sim \bar{r}^2/\ell^2$,

$$\Delta x_{far}/2 \sim \zeta \,, \quad \text{for} \quad \bar{r} \gg \ell \,, \tag{51}$$

which can be neglected compared to the throat contribution.

In the small $\bar{r}$ regime, $\mathcal{C} \approx 1$, so

$$\Delta x_{far}/2 \approx \frac{\pi\ell}{2L} \,, \quad \text{for} \quad \bar{r} \ll \ell \,. \tag{52}$$

Recall now that

$$L = \frac{\bar{r}^3 \pi \overline{\Delta x}^2}{Gq\lambda} \,, \tag{53}$$

so in the small $\bar{r}$ regime,

$$\Delta x_{far}/2 \approx \frac{G\ell q\lambda}{2\overline{\Delta x}^2 \bar{r}^3} \approx \frac{\ell \lambda g^3}{2\pi^{3/2} q^2 l_P \overline{\Delta x}^2} \,. \tag{54}$$

Recall that

$$\overline{\Delta x} = \frac{\Delta x_{throat} + \Delta x_{far}}{\pi} \geq 1 \,, \tag{55}$$

where the inequality arises due to the throat contribution. For $q^2 \gg \lambda \ell m_P g^3$ the far contribution is again small and can be neglected. However, for $q$ smaller than this value, this far contribution cannot be neglected.

We can now solve for $\overline{\Delta x}$ via

$$\overline{\Delta x} = 1 + \frac{\ell \lambda g^3}{\pi^{5/2} q^2 l_P \overline{\Delta x}^2} \,. \tag{56}$$

This is a cubic equation so the answer is unwieldy. For our purposes, it is sufficient to have the formula in the two limits:

$$\overline{\Delta x}^3 = \begin{cases} 1\,, & q^2 \gg \dfrac{\ell \lambda g^3}{l_P} \,, \\ \dfrac{\ell \lambda g^3}{\pi^{5/2} q^2 l_P}\,, & q^2 \ll \dfrac{\ell \lambda g^3}{l_P} \,. \end{cases} \tag{57}$$

The rough intuition behind this is straightforward: as the black hole becomes small, the total length of the wormhole is dominated eventually not by the near horizon region, but the asymptotic region. Alternatively, it is clear that the far region must eventually dominate if we fix all other length scales and take the AdS radius $\ell$ arbitrarily large. We do not have a quick argument for the particular scaling seen in the small $q$ regime.

### 2.1.7 Wormhole mass

The effect of the non-local coupling can be seen in the asymptotic mass, which is equal to the mass of the extremal RN-AdS black hole $M_{\text{ext}}$ together with a negative correction $\Delta M$. There are two contributions to the asymptotic correction to the mass $\Delta M$. The first can be read off from the matching, which determines the mass of the RNAdS black hole fits smoothly onto the throat solution. The result is

$$\boxed{\Delta M_{throat} = -\frac{\mathcal{C}(\bar{r})\bar{r}\pi^2\zeta^2}{32G} = -\frac{\mathcal{C}(\bar{r})\bar{r}^3}{2GL^2} = -\frac{\mathcal{C}q\lambda}{2\pi\overline{\Delta x}^2 L} \,.} \tag{58}$$

There is an additional contribution due to the Casimir energy in the asymptotic region, which becomes important for smaller black holes. The conserved Killing energy due to the Casimir energy in the far region is

$$E_{far}^{Cas} = \int d^3x \sqrt{g}\, T_{\mu\nu}\xi^\mu n^\nu = 2 \cdot 4\pi \int_{r_m}^{\infty} \frac{dr}{\sqrt{f}} r^2 T_{\tau\tau} \frac{1}{\sqrt{f}} = 8\pi \int_{r_m}^{\infty} dr\, r^2 T^\tau_{\ \tau} \,, \tag{59}$$

where the integral is over a constant time slice, $\xi^\mu$ is the timelike Killing vector corresponding to the asymptotic time $\tau$, and $n^\nu$ is the unit normal to the time slice.

In fact, although this is the natural quantity to compute from the perspective of QFT in a fixed background, once we backreact the solution it is more convenient to consider one of the definitions of mass that is guaranteed to agree with the ADM mass. A convenient choice is the so-called "$\rho$ mass", which is simply given by

$$M_\rho = 4\pi \int dr\, r^2 T^\tau_{\ \tau} \,. \tag{60}$$

This formula agrees with the above Killing energy in the region of interest.

Using the previous formula for $T_{tt}$ together with the relation $t = \mathcal{C}\tau/L$ gives

$$\langle T_{\tau\tau} \rangle = -\frac{q\lambda}{2\pi^3 \overline{\Delta x}^2 r^2 L^2} \,. \tag{61}$$

Since this correction is only important for small black holes we have set $\mathcal{C} = 1$. This gives

$$E_{far}^{Cas} = -\frac{2q\lambda}{\pi \overline{\Delta x}^2 L} \left( \frac{2}{\pi L} \int_{r_m}^{\infty} \frac{dr}{f} \right) \,. \tag{62}$$

The factor in parentheses is simply $\overline{\Delta x}_{far}$, so this contribution is

$$E_{far}^{Cas} = -\frac{2q\lambda \overline{\Delta x}_{far}}{\pi \overline{\Delta x}^2 L} \,. \tag{63}$$

We would like to translate this into a correction to $\Delta M$. There is a factor of 2 because the total energy shift is $2\Delta M$ due to the two asymptotic regions so that

$$\boxed{\Delta M_{far} = -\frac{q\lambda \overline{\Delta x}_{far}}{\pi \overline{\Delta x}^2 L} \,.} \tag{64}$$

Comparing the throat and far contributions, for large black holes the length in the far region is small, and also the mass shift from this region is negligible. However, in the regime where the far contribution $\overline{\Delta x}_{far}$ to the wormhole length dominates, $\Delta M$ is dominated by the far contribution.

Plugging in the value of $L$ gives

$$\Delta M \approx \begin{cases} -\dfrac{q\lambda}{\pi \ell}\,, & q^2 \ll \ell m_P \lambda g^3 \,, \\[2mm] -\dfrac{\mathcal{C} G q^2 \lambda^2}{2\pi^2 \bar{r}^3}\,, & q^2 \gg \ell m_P \lambda g^3 \,. \end{cases} \tag{65}$$

Note that $\bar{r}$ is determined by $q$.

One may want to further consider the limits within the larger charge regime above

$$\Delta M \approx \begin{cases} -\dfrac{\lambda}{\pi} \dfrac{q}{\ell}\,, & q^2 \ll \ell m_P \lambda g^3 \,, \\[2mm] -\dfrac{\lambda^2 g^3}{\pi^{7/2}} \dfrac{m_P}{q}\,, & q^2 \gg \ell m_P \lambda g^3 \,, \text{ and } \bar{r} \ll \ell \,, \\[2mm] -\dfrac{\lambda^2 \sqrt{g} 3^{5/4}}{\pi^{9/4}} \dfrac{l_P^{3/2} q^{3/2}}{\ell^{5/2}}\,, & q^2 \gg \ell m_P \lambda g^3 \,, \text{ and } \bar{r} \gg \ell \,. \end{cases} \tag{66}$$

This last case demonstrates the characteristic large charge behaviour identified by [21]. All in all, we thus have

$$M_{\text{WH}} = M_{\text{ext}} + \Delta M \,, \quad \text{where} \quad M_{\text{ext}} = \frac{\bar{r}}{G} + \frac{2\bar{r}^3}{G\ell^2} \,, \tag{67}$$

and $\Delta M$ as given in (65).

We end this subsection with some comments. First of all, the wormhole solution presented here is consistent with the Ansatz we made in (6). There are no horizons or singularities

anywhere in the solution. From outside of the wormhole throat, the solution is that of a super-extremal black hole, that naively has a naked singularity at its centre. Before one reaches that singularity, the geometry smoothly interpolates to a deformation of $AdS_2 \times S^2$, which has no horizons or singularities. Second, we comment on the number of parameters. There are three independent parameters that determine the wormhole solution uniquely: the AdS length $\ell$, the charge $q$, and the non-local coupling $h$.[6] As soon as these parameters are fixed the solution presented is the unique static spherically symmetric solution. Third, note that the deformations $\psi$ and $\gamma$ break Poincaré invariance. Therefore, our solution evades the no-go result of [22].

## 2.2 Alternative derivation of the Casimir energy

In this subsection, we show an alternative derivation of the energy difference between the extremal RN solution and our wormhole, by summing over the frequencies. This shows that we could have found the Casimir energy without evaluating the backreaction on the geometry.

The effect of the non-local coupling manifests itself in the shift of the frequencies $\omega_k$ of the fermionic modes.[7] Moreover, the quantized fermionic field gives rise to a fermionic harmonic oscillator, whose energy is given by minus half the sum over the frequencies. This infinite sum over frequencies formally diverges. However, we know that for $h = 0$, the regularized sum over frequencies results in the total energy of the extremal RN black hole. We can thus evaluate the sum over $\Delta\omega_k := \omega_k - \omega_k^{h=0}$, and find the energy difference between the wormhole and the extremal RN black hole. Using (17), we see that

$$\Delta E = -q \left( \sum_{k \geq 0} \Delta\omega_k - \sum_{k < 0} \Delta\omega_k \right) = -q\Delta\omega_0 = -\frac{2q\lambda(h)}{\pi\overline{\Delta x}} \,. \tag{68}$$

Here the factor $q$ comes from the fact that in the lowest Landau level approximation, we have $q$ (complex) fermionic harmonic oscillators. Note that this is the energy difference with respect to the Killing vector $\partial_t$ in the metric (6). We can relate it to the asymptotic metric (27) by the coordinate transformation

$$t = \frac{\tau}{L}\mathcal{C}(\bar{r}), \quad x = \int dr \frac{1}{L}\frac{1}{f(r)}\mathcal{C}(\bar{r}), \tag{69}$$

and by setting

$$e^{\sigma(x)} = \frac{L\sqrt{f(r)}}{\mathcal{C}(\bar{r})} \,. \tag{70}$$

Therefore, in the asymptotic coordinates, we find the energy difference to be

$$\Delta E_\tau = -\frac{\mathcal{C}(\bar{r})}{L}\frac{2q\lambda(h)}{\pi\overline{\Delta x}} \,. \tag{71}$$

This energy difference agrees exactly with (63) in the limit $\overline{\Delta x} \approx \overline{\Delta x}_{far}$. Evaluating the sum over frequencies is thus a much faster derivation for $\Delta M$ in this limit. For small black holes, it agrees with the total $\Delta M$, while for large black holes, the sum over frequencies cannot account for the total $\Delta M$ due to the backreaction effects that are not captured in this purely QFT calculation; it only accounts for the difference in Casimir energy in the fixed background, and not for the gravitational effects that are important in the throat region.

---

[6]Note that for the construction to work we must have that $q$ is large and $\zeta$ small. $q$ has to be large to be able to restrict to the lowest Landau level, while $\zeta$ must be small for the linearized Einstein equations to be a good approximation.

[7]Notice that in our conventions, the frequencies are given by $|\omega_k|$, and we have modes for all $k \in \mathbb{Z}$.

We end with a comment: the computation in this subsection, although much simpler, is not enough to show that there exists a traversable wormhole solution. To ensure that the solution is in fact traversable, one has to show that the backreacted geometry, sourced by the stress tensor computed in the Ansatz (6) is self-consistent. In other words, one needs to show that the backreacted geometry can be written in the form (6), without any horizons or singularities, as demonstrated in the previous subsection. Moreover, we also need the gravity analysis to determine the correct value of $\overline{\Delta x}$, allowing us to solve for the energy in terms of the charge.

## 2.3 Semiclassical breakdown for long wormholes

The traversable wormhole has a maximum length for the semiclassical approximation to be valid; this can be argued through their similarity to nearly-extremal RN black holes [23]. The semiclassical approximation of RN black holes breaks down at sufficiently low temperatures because the average energy of Hawking quanta becomes the same order as the mass of the black hole above extremality, so fluctuations in the near horizon geometry cannot be neglected [24]. The difference in mass between a nearly extremal RN black hole and an extremal one of the same charge is

$$M - M_{ext.} = M_{gap}^{-1} T^2 + O(T^3),\tag{72}$$

where

$$M_{gap}^{-1} = \frac{2\pi^2 \bar{r}^3}{G\mathcal{C}(\bar{r})}.\tag{73}$$

When $T M_{gap}^{-1} \lesssim 1$, (72) is less than the average energy of Hawking quanta, which is $\langle E \rangle \sim T$. Thus, to be in the semiclassical regime, requires

$$T \gg \frac{G\mathcal{C}(\bar{r})}{2\pi^2 \bar{r}^3}.\tag{74}$$

The lower bound on temperature places an upper bound on the length of the near-horizon throat region of the near-extremal black hole, through the relation between throat length and RN black hole temperature, given by

$$L \ll \frac{q^3}{m_p g^3}.\tag{75}$$

The length of our wormhole is given by (53), which for large $q$ is

$$L = \frac{\pi m_p^2 \bar{r}^3}{q\lambda(h)}.\tag{76}$$

This, in combination with (75), places a lower bound on our boundary coupling $h$. The form of the bound on $h$ depends on our input parameters. For $r_e \ll \ell$ the bound is $h \gg q^{-1}$. If the boundary coupling $h$ is too small, then our wormhole is too long for the semiclassical approximation to be valid.

## 3 Phases

In this section, we will revisit and expand on some results first presented in section 4.1 of [9]. We clear up a subtlety that went unnoticed and generalize the discussion to finite non-local coupling. We also discuss the effects of working in either the canonical or microcanonical ensemble and the possibility of fragmentation.

### 3.1 Boundary Hamiltonian

In the boundary theory, the dual of the wormhole solution is some highly entangled state. Moreover, this entangled state is expected to be the ground state of a local Hamiltonian for some range of the parameters [25, 26]. In the bulk theory, given boundary conditions, gravity fills in the geometry smoothly. We will consider three possible gravitational solutions at zero temperature: Empty AdS, two disconnected *extremal* RN black holes, and the wormhole solution. To simplify matters, we will focus on the setup that has some additional symmetry between the left and right sides: the magnetic charge is $Q := Q_R = -Q_L = q/g$, and the mass is $M = M_L = M_R$. Our goal is then to propose a boundary Hamiltonian, for which our traversable wormhole is the ground state in some region of parameter space.

We start by computing the on-shell Hamiltonian of the traversable wormhole solution presented in section 2. The on-shell Hamiltonian is given by the charge associated with a symmetry given by a boundary Killing vector $\zeta$, which can be computed using the Brown-York stress tensor as

$$H[\zeta] = \int_{\partial\Sigma} d^2x \sqrt{\sigma} u^\alpha \zeta^\beta T_{\alpha\beta}, \quad \text{where} \quad T_{\alpha\beta} := \frac{2}{\sqrt{-\gamma}} \frac{\delta S}{\delta \gamma^{\alpha\beta}}, \tag{77}$$

where $\Sigma$ is a constant timeslice, $u$ is the unit normal to it, and $\sqrt{\sigma}$ is volume element at its boundary $\partial\Sigma$. We will be interested in the gravitational energy, which is given by the charge associated with symmetry under time-translations. In other words, we take $\zeta = \partial_\tau$.

In [9], (77) was applied to the interacting part of the action (15). However, this is not the correct way to think about the effect of the non-local interaction on the on-shell Hamiltonian. The manner in which the on-shell Hamiltonian knows about the non-local interaction is through the backreacted geometry [27]. This has been known for "ordinary matter", but the matter action also does not contribute to the on-shell Hamiltonian when including our non-local boundary action. To see this we first note that from (77) it follows that only the boundary contributions to any matter action can contribute to the on-shell Hamiltonian. One should however be mindful of boundary terms coming from the bulk matter action. We will show that once we combine all boundary terms associated with the fermionic field $\Psi$, the on-shell action vanishes due to the boundary condition on $\Psi$, given in (16). To do so we first need to consider the bulk action. As is well-known, the action is proportional to the equations of motion. In the presence of boundaries, however, there are terms localized on the boundary that one should be careful about. To see this we rewrite the bulk matter action as bulk terms that vanish on-shell and some boundary terms that do not by themselves vanish:[8]

$$S_M = -\int d^4x \sqrt{g} \left( \psi_+^\dagger \partial_x \psi_+ + \psi_+^\dagger \partial_t \psi_+ - \psi_-^\dagger \partial_x \psi_- + \psi_-^\dagger \partial_t \psi_- \right) \eta^\dagger \eta \tag{78}$$

$$= -\int d^4x \sqrt{x} \Big[ \partial_x \left( \psi_+^\dagger \psi_+ - \psi_-^\dagger \psi_- \right) + \partial_t \overset{0}{\cancel{\left( \psi_+^\dagger \psi_+ + \psi_-^\dagger \psi_- \right)}} \tag{79}$$

$$\underset{}{\cancel{-\partial_x \psi_+^\dagger \psi_+ + \partial_x \psi_-^\dagger \psi_- - \partial_t \psi_+^\dagger \psi_+ - \partial_t \psi_-^\dagger \psi_-}}^{\text{e.o.m}} \Big] \eta^\dagger \eta$$

$$= -\int_\partial d^3x \sqrt{\gamma} \left( \psi_+^\dagger \psi_+ - \psi_-^\dagger \psi_- \right) \eta^\dagger \eta \tag{80}$$

$$= \int_\partial d^3x \sqrt{\gamma} \left( \overline{\Psi}_- \Psi_+ - \overline{\Psi}_+ \Psi_- \right). \tag{81}$$

---

[8]We restrict ourselves to the lowest Landau level.

Note that here $\eta = \begin{pmatrix} \eta_- \\ \eta_+ \end{pmatrix}$, with $\eta_+ = 0$, and $\eta_- = \sum_m C_m^j \eta_-^m$, and $\psi_\pm$ given by the sum over modes given in (9). Combining this with the boundary action as given in (13), we find

$$S_\partial^{\text{total}} = \int_\partial d^3x \sqrt{\gamma} \Big[ \underbrace{\overline{\Psi}_-^R \Psi_+^R - \overline{\Psi}_+^R \Psi_-^R - \overline{\Psi}_-^L \Psi_+^L + \overline{\Psi}_+^L \Psi_-^L}_{\text{bulk}} + \underbrace{\overline{\Psi}_+^R \Psi_-^R + \overline{\Psi}_-^L \Psi_+^L}_{S_{\text{classical}}^\partial} + \underbrace{h\overline{\Psi}_-^R \Psi_+^L + h\overline{\Psi}_+^L \Psi_-^R}_{S_{\text{non-local}}^\partial} \Big]$$

(82)

$$= \int_\partial d^3x \sqrt{\gamma} \Big[ \overline{\Psi}_-^R \left( \Psi_+^R + h\Psi_+^L \right) + \overline{\Psi}_+^L \left( \Psi_-^L + h\Psi_-^R \right) \Big]$$

(83)

$$= 0 \,,$$

(84)

due to the boundary condition. Therefore, the matter action does not contribute to the on-shell Hamiltonian, even in the presence of non-standard boundary conditions.

In four dimensions, the Brown-York stress tensor is given by

$$T_{\alpha\beta} = K_{\alpha\beta} - K\gamma_{\alpha\beta} - \frac{2}{\ell}\gamma_{\alpha\beta} - \ell G_{\alpha\beta} \,,$$

(85)

where $G_{\alpha\beta}$ is the Einstein tensor due to the boundary metric $\gamma_{\alpha\beta}$, and we have included the contribution of the local counterterm that is needed to regulate IR divergences.

At the asymptotic boundary, the timelike unit vector is given by $u = \sqrt{f(r)}d\tau$. It is then a simple exercise to show that the expectation value of the on-shell Hamiltonian is given by the asymptotic gravitational energy

$$\langle H \rangle_{\text{WH}} = 2M_{\text{WH}} \,.$$

(86)

In the above equation, the factor of two appears due to the fact that we integrate (77) over the entire boundary *i.e.* both the left and right boundary.

Inspired by this result, we can define a *local* boundary Hamiltonian. First, we define the boundary spinors $\Psi_\pm^{R,L}$, which are related to the bulk spinors $\Psi_\pm$ in the following way

$$\Psi_\pm^{R,L} = R^{\frac{3}{2}} \Psi_\pm^{R,L} \,.$$

(87)

In terms of the boundary data, the local boundary Hamiltonian we propose is given by

$$H := H_L + H_R - \frac{ih}{\ell} \int d\Omega_2 \left( \overline{\Psi}_-^R \Psi_+^L + \overline{\Psi}_+^L \Psi_-^R \right) + \mu(Q_L - Q_R) \,.$$

(88)

Here $H_L$ and $H_R$ should be interpreted as the local Hamiltonians associated with two identical systems of the boundary theory without the non-local interaction, and $Q_R = -Q_L = q/g$. This Hamiltonian governs the time-evolution with respect to the asymptotic time $\tau$, and is conserved due to time-translation symmetry.

We propose this particular Hamiltonian because, for the wormhole solution, the terms independent of the chemical potential evaluate precisely to (86) by construction. For the phase consisting of two disconnected extremal RN black holes the interacting part of the Hamiltonian does not explicitly contribute for the same reason as in the wormhole phase and the expectation value of the Hamiltonian is given by

$$\langle H \rangle_{\text{2BH}} = 2M_{\text{ext}} - 2\mu Q \,,$$

(89)

with $M_{\text{ext}}$ the mass of the extremal black hole of charge $Q$. Finally, for the empty AdS phase, the Hamiltonian vanishes.

## 3.2 Identifying the ground state

Since $\Delta M < 0$, we always have that

$$\langle H \rangle_{\text{2BH}} - \langle H \rangle_{\text{WH}} > 0 \,. \tag{90}$$

It is then easy to see that $\langle H \rangle_{\text{2BH}}$ is minimized by

$$\bar{r} = \frac{\ell}{\sqrt{3\pi}} \sqrt{\frac{\mu^2}{m_p^2} - \pi} \,, \quad \text{for} \quad \frac{\mu^2}{m_p^2} > \pi \,. \tag{91}$$

Moreover, for these values $\langle H \rangle_{\text{2BH}} < 0$, and hence both the black hole and the wormhole phase have lower energy than empty AdS. From this, we can conclude that for the values $\mu > \mu_c$, with

$$\mu_c = m_p \sqrt{\pi} \,, \tag{92}$$

the wormhole phase is the ground state whenever it exists.

One may wonder whether once $h$ is turned on, the wormhole can be the ground state even when $\mu < \mu_c$. We cannot analytically solve the equation needed to answer this question precisely, which is

$$\langle H \rangle_{\text{WH}} < 0 \,. \tag{93}$$

We can however infer some information about the phase transition between the wormhole and empty AdS phases using symmetry, by investigating the scales involved and we can solve in an appropriate approximation. First, $\langle H \rangle_{\text{WH}}$ only depends on the non-local coupling $h$ through $\lambda(h)$, which is symmetric under $h \mapsto h^{-1}$. Therefore, the phase diagram should be symmetric under this transformation. Second, we can argue that there is no value of $h$ for which the wormhole phase extends all the way to $\mu \to 0$. We can argue that this is not the case by solving (93) for $\mu = 0$. This leads to the following inequality

$$\frac{\bar{r}^2}{G\sqrt{q}} \lesssim \lambda(h) \,, \tag{94}$$

where we have ignored order one numbers. By noting that $\lambda(h) \leq \frac{\pi}{4}$, it follows that (93) is never satisfied at very small $\mu$ (while at the same time being in the semiclassical regime). Even though we cannot solve (93) analytically, we can do perturbation theory around $\mu = \mu_c$. For such $\mu$ it follows from (91) that $\bar{r}$ is small compared to $\ell$ in this regime. Therefore, solutions of (93) are well approximated by solutions to the same equation with

$$M_{\text{ext}} = \frac{\bar{r}}{G} \,, \quad \Delta M = -\frac{\lambda(h)q}{\pi\ell} \,, \quad \text{and} \quad r_e^2 = \bar{r}^2 \,. \tag{95}$$

In this approximation, we find that (93) is solved by

$$\boxed{\mu(h) > \mu_c \left( 1 - \frac{g\lambda(h)l_P}{\pi^{3/2}\ell} \right) \,.} \tag{96}$$

First of all, we note that this bound is consistent with our aforementioned arguments. Also, notice that the correction to $\mu_c$ is very small for typical parameter values.

Using the results we infer that the phase diagram is approximately as shown in figure 1. Moreover, near the transition, we have

$$\langle H/2 \rangle \approx -\left( \frac{2}{3\sqrt{\pi}} \right)^{3/2} \frac{\ell}{\sqrt{l_P}} (\mu - \mu_c)^{3/2} \,. \tag{97}$$

### 3.3 Ensembles and when the traversable wormhole dominates

The question of when the traversable wormhole is the dominant bulk configuration can be asked in various ensembles. Just like for small black holes in asymptotically AdS spacetime, interesting differences can arise between different ensembles. Recall that in asymptotically AdS$_5$ spacetime, small black holes (with a radius less than the AdS radius) never dominate the canonical ensemble, but they do dominate the microcanonical ensemble for a wide range of masses [28].

Before coupling the two CFTs there are two conserved $U(1)$ charges, but after coupling only the total charge $Q_L + Q_R$ remains. One can consider turning on a chemical potential for this conserved charge, but in the following, we will simply fix $Q_L + Q_R = 0$ for simplicity.

#### 3.3.1 Microcanonical ensemble

We have seen above the region of parameters $(\mu, h)$ for which the traversable wormhole is the ground state. Now consider higher energy states. In the microcanonical ensemble, the dominant state is determined by the state with the largest entropy at the given energy. We work in the limit of small $G$ so that the only entropy we consider is horizon entropy.

Consider the region of parameters where the ground state is the traversable wormhole. For energies in the range

$$E < 2M_{\text{ext}} - 2\mu Q, \tag{98}$$

the traversable wormhole is the only solution. (Recall that in this regime of parameters, the above energy is negative, so empty AdS is also not an allowed solution.) The relevant solution is a 'hot' traversable wormhole: a traversable wormhole with some additional thermal radiation to raise the energy above the ground state energy.

Now consider the range of energies

$$E > 2M_{\text{ext}} - 2\mu Q. \tag{99}$$

In this regime, both the hot traversable wormhole and the black hole are allowed solutions. (At some energy the hot traversable wormhole will become unstable due to the backreaction, but we expect this to happen at higher energy than where it stops dominating the ensemble. It would be interesting to check this.) To the order we are working to in the $G$ expansion, the hot traversable wormhole has zero entropy, while the black hole has a significant horizon entropy. Therefore, in the full regime where the black hole solution exists, it dominates the microcanonical ensemble.

Empty AdS does not dominate at any energy in this range of parameters, because the black hole is a possible solution for all $E > 0$ and has larger entropy than empty AdS. To recall, this happens because the chemical potential-like term in our Hamiltonian shifts the conserved energy from $2M$ to $2(M - \mu Q)$, where $M$ is the asymptotic mass as measured by the metric.

#### 3.3.2 Canonical ensemble

To see which phase dominates the canonical ensemble, we need to compute the free energy $F = E - TS$. We again work in the purple regime of parameters where the traversable wormhole is the ground state. The free energy of the traversable wormhole is

$$F_{TWH} = 2(M_{\text{ext}} + \Delta M - \mu Q). \tag{100}$$

Recall that $M_{\text{ext}}$ is determined by $Q$ and $\Delta M$ is determined by $Q$ and the coupling $h$, so this result is *independent* of the temperature.

The free energy of the black hole solution is

$$F_{2BH} = 2\left(M(Q,T) - \mu Q - TS(Q,T)\right), \tag{101}$$

where $M(Q,T)$ is the mass of a black hole with charge $Q$ and temperature $T$. Due to the entropy term, this solution will begin to dominate at low temperatures. At low temperatures, the free energy takes the simpler form

$$F_{2BH} \approx 2\left(M_{\text{ext}}(Q) - \mu Q - TS_{\text{ext}}(Q)\right). \tag{102}$$

Like in the microcanonical case discussed above, in this range of parameters, the dominant phase is always the traversable wormhole or the black holes. Comparing the free energies of these two solutions, we find that the black hole begins to dominate at the temperature

$$T_c = \frac{\Delta M}{S_{\text{ext}}(Q)}. \tag{103}$$

This transition temperature is lower than the temperature (73) at which the semiclassical description of the black hole breaks down; $T_c$ larger than (73) is inconsistent with $g \ll 1$. For example, for a small $\bar{r}$, $\Delta M$ is given by

$$\Delta M \sim -\frac{g^2 \lambda^2}{\bar{r}}, \tag{104}$$

so that its magnitude is bounded by

$$|\Delta M| \lesssim \frac{1}{\bar{r}}, \tag{105}$$

and the critical temperature is bounded by

$$T_c \lesssim \frac{1}{\bar{r} S_{\text{ext}}}. \tag{106}$$

Our energy gap $\Delta M$ is not particularly small, as can be seen from the above equation. However, the fact that the black hole solution begins to dominate at very low temperatures may indicate difficulties in preparing the traversable wormhole state.

The situation is similar to the small black holes in AdS mentioned above. In the microcanonical ensemble, the traversable wormhole dominates for a range of energies up to $\Delta M$ above the ground state, and its temperature can be as high as $T \sim \Delta M$. But in the canonical ensemble, these hot traversable wormholes never dominate, except at extremely low temperatures.

To summarize: If one can prepare the system in a particular range of energies, it is straightforward to choose a range of parameters where the traversable wormhole dominates, but if one can only fix the temperature preparation will be difficult. While it may be more natural to do experiments at a fixed temperature, it is easier to do simulations at fixed energy. Therefore, this issue does not pose a problem for simulating traversable wormholes.

## 3.4 Fragmentation

Is the true ground state in the wormhole phase a single large wormhole, or a collection of small wormholes?

Results on AdS fragmentation [29] and holographic vitrification [30] indicate that near-extremal black holes are able to fragment. In the above-mentioned cases, often entropy considerations make the un-fragmented black hole the dominant configuration. However, our

wormholes do not have a large entropy, so the dominant configuration will be determined by energetic considerations.

We do not keep track of order one-factors in this section. For simplicity, we also neglect the gauge coupling $g$ and the strength of the Casimir interaction $\lambda$,

$$g \sim \lambda \sim 1\,. \tag{107}$$

Consider a configuration with a total charge of $q$. We first work with small black holes with a horizon size much smaller than the AdS radius. Calculations in the previous sections of the paper give

$$\Delta M \sim -\frac{Gq^2}{r_e^3} \sim -\frac{m_P}{q}\,. \tag{108}$$

The fact that the energy is inversely proportional to $q$ suggests that fragmentation is energetically favoured. Naively, two wormholes with charge $q/2$ would have more negative energy than a single wormhole with charge $q$. We want to compute whether this naive expectation is correct.

We first need a better formula for the length $L_w$ of the wormhole, applicable also for very small wormholes. Recall that

$$L_w \sim L \sim \frac{\bar{r}}{\epsilon}\,, \tag{109}$$

which is computed under the assumption that most of the wormhole length comes from the near-horizon region. A more general formula is

$$L_w \sim L + \ell \sim \frac{\bar{r}}{\epsilon} + \ell\,, \tag{110}$$

which simply adds the distance in the external region. This can be justified further via conformally mapping the metric. Using

$$\epsilon^2 \sim -\frac{G\Delta M}{\bar{r}}\,, \tag{111}$$

we have

$$L_w \sim \frac{\bar{r}^{3/2}}{l_P(-\Delta M)^{1/2}} + \ell \sim \frac{q^{3/2}l_P^{1/2}}{(-\Delta M)^{1/2}} + \ell\,. \tag{112}$$

We can solve for $\Delta M$ by combining the above equation with

$$E_{cas} = \Delta M = -\frac{q}{L_w}\,. \tag{113}$$

Solving for $\Delta M$ gives the messy formula

$$\ell\Delta M = q + 2\frac{l_P}{\ell}q^3 - 2\sqrt{\frac{l_P}{\ell}q^{3/2}}\sqrt{\frac{l_P}{\ell}q^3 + q}\,. \tag{114}$$

This simplifies in limits:

$$\ell\Delta M = \begin{cases} -q + 2\sqrt{\dfrac{l_P}{\ell}}q^2 + \cdots, & q^2 \ll \dfrac{\ell}{l_P}\,, \\[3mm] -\dfrac{\ell}{l_P q} + \cdots, & q^2 \gg \dfrac{\ell}{l_P}\,. \end{cases} \tag{115}$$

Now we want to consider breaking up the total charge $q$ into $N$ wormholes, each with charge $q_1 = q/N$. If these wormholes are not too close together, the Casimir contribution to each wormhole can be computed independently and is given by the formula above, with

$q$ given by the charge of the individual wormhole. If we only take into account the Casimir energy, we have a total energy

$$\ell \Delta M_{tot} = \begin{cases} -q + 2\sqrt{\dfrac{l_P}{\ell}} q^2/N + \cdots, & \left(\dfrac{q}{N}\right)^2 \ll \dfrac{\ell}{l_P}, \\ -\dfrac{\ell N^2}{l_P q} + \cdots, & \left(\dfrac{q}{N}\right)^2 \gg \dfrac{\ell}{l_P}. \end{cases} \tag{116}$$

As long as $N$ is large enough so that we are in the upper case, this is minimized by making a very large number of wormholes, $N \to \infty$; the total energy is independent of $N$ for large $N$ and is simply

$$\Delta M \sim -\frac{q}{\ell}. \tag{117}$$

Considering smaller $N$ so that we are in the lower case, we want to choose the minimal $N$, giving $N^2 = q^2 l_P/\ell$ so that

$$\Delta M_{tot} \sim -\frac{q}{\ell}, \tag{118}$$

which agrees with the previous formula. To the accuracy that we are computing, this agrees with the formula for a single wormhole in the small $q$ regime. It is more negative than the single wormhole formula in the large $q$ regime.

However, so far we have neglected the interaction between the wormholes. We want to now estimate the corrections. There are two types of corrections: the wormholes interact with each other gravitationally and through electromagnetic interactions, and experience the AdS gravitational potential. A more careful analysis is possible, but for now, we assume the wormholes are in a region much smaller than the AdS radius so that we can use flat space formulae. In that case the interactions between the wormholes nearly cancel because they are near extremal. The correction is due to $\Delta M$,

$$E_{int} \sim -\sum_{ij} \frac{GM_i \Delta M_j}{|r_{ij}|}, \tag{119}$$

where the sum goes over all wormholes. The interactions of the nearby wormholes do not dominate, so if the wormholes are spread over a region of size $d$ the interaction energy is approximately

$$E_{int} \sim -N^2 \frac{GM_1 \Delta M_1}{d}. \tag{120}$$

Using $N\Delta M_1 = \Delta M_{tot}$ and $M_1 = q_1 l_P^{-1}$ gives

$$E_{int} \sim -\frac{\Delta M_{tot} l_P q}{d} \sim \frac{q^2 l_P}{d\ell}. \tag{121}$$

Note that the factors of $N$ have cancelled each other, so this contribution does not care how much we fragment the black hole, within our approximation here.

The AdS potential contributes

$$E_{AdS} \sim \sum_i M_i \Phi(x_i) \sim NM_1 \frac{d^2}{\ell^2} \sim \frac{qd^2}{\ell^2 l_P}. \tag{122}$$

Again the factor $N$ has cancelled out of the answer.

Optimizing $d$ gives

$$d^3 \sim q\ell l_P^2 = \ell^3 \frac{q l_P^2}{\ell^2}. \tag{123}$$

The above formula is valid as long as $d < \ell$, requiring $q < \ell^2/l_P^2$. This is already implied since we are considering small black holes with $\bar{r} \sim l_P q \ll \ell$.

An additional bound on $d$ is that the separation between the wormholes has to be larger than their Schwarzschild radii. For the above formulas to be valid, the separation should be much larger,

$$\frac{d^3}{N} \gg \bar{r}^3 \sim \left(\frac{ql_P}{N}\right)^3 \implies d \gg \frac{ql_P}{N^{2/3}}. \tag{124}$$

Additionally, the whole collection of wormholes should be outside its Schwarzschild radius,

$$\frac{Gq}{dl_P} \ll 1 \implies d \gg ql_P, \tag{125}$$

which is a stronger requirement than the previous one. This is satisfied by the optimal $d$ if

$$q\ell l_P^2 \gg q^3 l_P^3, \quad \text{and} \quad q \ll \sqrt{\frac{\ell}{l_P}}. \tag{126}$$

But this (to the accuracy we are working) implies that we are in the small $q$ regime, where the single wormhole has the same energy as the fragmented wormhole. For larger charges, our fragmented analysis breaks down.

Plugging in the optimal $d$ gives the energy of the configuration,

$$E_{non-cas} \sim q^{5/3} l_P^{1/3} \ell^{-4/3} \sim |E_{cas}| \left(\frac{q^2 l_P}{\ell}\right)^{1/3}. \tag{127}$$

To summarize: when the total charge is small enough, $q \lesssim \sqrt{\ell/l_P}$, the wormhole can fragment into a large number $N$ of small wormholes, with a total energy shift

$$\Delta M \approx E_{cas} \left[1 - \left(\frac{q^2 l_P}{\ell}\right)^{1/3} + \cdots\right], \quad \text{with} \quad E_{cas} \sim -\frac{q}{\ell}. \tag{128}$$

*The mass shift $\Delta M$ is independent of the number of fragments to the order we have analyzed it.* Therefore, we cannot conclude from this analysis whether wormholes in this range of charges want to fragment or not. In computing the phase diagram, we can simply use the single wormhole formulas for the energy of the configuration. Fragmentation could correct the entropy of the configuration, but since we only include terms of order $G^{-1}$ in the entropy, this correction can be neglected. The much more detailed results of [30] indicate that stable fragmented configurations in AdS require the fragments to have mutually non-local charges; therefore, our simple configuration with purely magnetic charges would not be expected to be stable.

For a larger total charge, the above analysis breaks down as we described, but the natural expectation is that larger wormholes do not want to fragment. This can be checked by computing the dynamics of a small wormhole outside a larger wormhole. Treating the small wormhole as a probe point particle, the action is

$$S_{probe} = -m \int d\tau - e \int A, \tag{129}$$

where $m$ is the mass of the probe, $e$ is its charge, and the integral is over the worldline of the particle. The wormhole is slightly super-extremal,

$$m = em_P + \delta m, \tag{130}$$

with $\delta m < 0$. This small correction will prove inconsequential.

Considering static configurations, the probe action is

$$S_{static} = -\int dt\left(m\sqrt{f(r)} + \frac{eq}{r}\right). \tag{131}$$

We identify the integrand with a potential $V(r)$. For a large AdS radius, compared to all other scales, the large $r$ behaviour is

$$V(r) \approx -\frac{GMm}{r} + \frac{eq}{r}, \tag{132}$$

simply the usual gravitational and electrostatic potentials. Since both the large wormhole and the probe are super-extremal, the probe is pushed towards infinity, indicating a tendency to fragment.

Now, however, consider the effect of the AdS radius. The effective potential,

$$V = m\sqrt{f} + \frac{eq}{r}, \tag{133}$$

increases at large $r$ as $V \approx mr/\ell$. For large black holes, $\bar{r} \gg \ell$, one can check that the potential is monotonic, so fragments are pushed back into the wormhole. Thus fragmentation is irrelevant for larger black holes.

## 3.5 Other potential instabilities

Besides the stability of the traversable wormhole, one might also worry about the stability of the near-extremal black hole phase, which we envision as the starting point of our tunneling experiment. It is well-known that AdS-RN black holes develop instabilities once they approach zero temperature and become extremal, see for example [31–37]. The experiment we have in mind is conducted either at fixed non-zero (but low) temperatures or at fixed energy.

An important question to ask is whether the decay channel to the wormhole phase is the dominant one. As mentioned in section 3.4, near extremal black holes can fragment, however, due to entropy considerations, the un-fragmented black hole is expected to be the dominant one. One could also worry about stability under Schwinger pair production of magnetic monopoles. We expect that one can make the mass of the monopoles sufficiently large such that the corresponding decay channel is a subleading effect. The Schwinger decay rate is exponentially suppressed by the mass of the monopole, and heavy monopoles are confined by the effective radial potential.

One further decay channel of concern is Hawking radiation because of its enhancement for near-extremal magnetic black holes due to the large degeneracy in the lowest Landau level [38]. From the considerations of section 3.3 it follows that for the region of parameters of interest, the black hole and wormhole phase dominate over empty AdS in either ensemble. Hence, instabilities concerning Hawking radiation are not relevant for our computations. The underlying reason is that in asymptotically AdS spacetimes, Hawking radiation is in thermal equilibrium with incoming thermal radiation.

# 4 Recipe for computing decay rates

In this section, we provide the recipe that we will use to compute decay rates. First we discuss the different choices for ensembles and boundary conditions in subsection 4.1, after which we discuss the example of emitting Reissner-Nordström black holes in 4.2 where we show that the recipe leads to sensible decay rates.

Instanton methods were originally formulated in flat non-dynamical spacetimes [39, 40], but have since been incorporated into gravitational theories [41–44], and used to study instabilities of spacetime [45–48]. Instantons have also been generalised to finite temperatures where they are also known as thermalons or calorons [49, 50]. Instantons and domain walls have also played a prominent role in cosmology [51–59].

## 4.1 Ensembles and boundary terms

In computing the decay rate via instantons, it matters what we hold fixed at the boundary. We can ask two different physical questions: what is the decay rate at fixed energy (microcanonical) or at fixed temperature? The decay rates for different ensembles may coincide in some situations, for example, due to ensemble equivalence, but in general, they can be different. We are familiar with examples where certain black holes are stable in the microcanonical ensemble but unstable in the canonical ensemble, for example for certain small black holes in AdS. So clearly gravity admits situations where the two ensembles are quite different.

The recipe for computing the decay rate is to compute the Euclidean action for the instanton and subtract it from the Euclidean action for the background solution. The appropriate instanton solution will depend on what is held fixed at the boundary. In addition, the action will depend on which boundary terms are present in the action, which of course is correlated to what quantity is held fixed.

### 4.1.1 Canonical ensemble

The most familiar formulation is the canonical ensemble. In Euclidean, this corresponds to fixing the metric at infinity. The appropriate action with these boundary conditions includes the Gibbons Hawking York boundary term,

$$
S_{\text{Euc}} = \int_M d^4x \sqrt{g} \left[ -\frac{R}{16\pi G} + \mathcal{L}_m \right] - \int_{\partial M} \sqrt{h} \frac{K}{8\pi G} \,, \tag{134}
$$

where $K$ is the trace of the extrinsic curvature of the boundary and $h$ is the induced metric of the boundary. There are differing sign conventions: this is the correct sign convention if the extrinsic curvature is defined with respect to an outward pointing normal at the boundary and $R$ is defined so that spheres have positive curvature. In rather general situations, the Euclidean matter Lagrangian is equal to the (Euclidean) energy density, $\mathcal{L}_m = T_{00}$.

When using this formulation, our instanton must have the same asymptotic values of the metric at infinity; the temperature is fixed. This means that in general, the mass at infinity for the instanton is different from the asymptotic mass of the background. This appears to happen in relatively simple situations, such as the emission of neutral particles by Reissner-Nordström black holes. There is nothing wrong with this: In the canonical ensemble, the energy is not fixed. However, if we are interested in a physical situation where the energy is fixed, we need to use a different formalism.

### 4.1.2 Microcanonical ensemble

In many physical situations, the total energy is conserved, so the microcanonical ensemble is more relevant. Microcanonical ensembles in AdS/CFT have previously been studied to understand the bulk physics of black holes at temperatures below the Hawking-Page transition [60]. One option is to not fix the asymptotic mass but to fix the induced metric on the sphere. This is treating the Euclidean time and space directions on unequal footing but is a physically sensible question to ask. Brown and York [61] defined a suitable action for these boundary conditions,

given in our current notation and conventions by

$$S_{\text{Euc}} = \int_M d^4x \sqrt{g}\left[-\frac{R}{16\pi G} + \mathcal{L}_m\right] - \int_{\partial M} \sqrt{h}\frac{t_\mu K^{\mu\nu}\partial_\nu t}{8\pi G}\,, \tag{135}$$

where "$t$ is the scalar field on the boundary that labels the foliation, ... and $t_\mu$ is the time vector field" defined on the boundary.

This formula is written by Iyer and Wald [62] as

$$I = \int_M \mathbf{L} - \int_{\partial M} dt \wedge \mathbf{Q}\,, \tag{136}$$

where $\mathbf{L}$ is the bulk Lagrangian density and $\mathbf{Q}$ is the Noether charge 2-form.

### 4.1.3 Alternative boundary conditions

**Fixing all normal derivatives.** The above proposal treats space and time differently. It is also possible to fix only the (suitably defined) normal derivatives of the metric and let the boundary metric fluctuate. In this case, the appropriate action, in 3+1 dimensions, has no boundary term. In general dimensions, the needed boundary term is a multiple of the GHY term.

**Fixing the conformal data of the boundary metric and $K$.** Witten [63] points out that gravitational perturbation theory is more well-behaved if, instead of fixing the full boundary metric, we fix only the conformal data of the boundary metric, and fix the trace of the extrinsic curvature of the boundary metric. The needed boundary term is a multiple of the GHY term,

$$S_\partial^W = \frac{1}{D-1}S_\partial^{GHY}\,. \tag{137}$$

## 4.2 Emission of thin radiation shells from Reissner-Nordström black holes

To test that we get sensible answers from this Euclidean procedure, we compute the emission of neutral radiation by a Reissner-Nordström black hole. For previous work on AdS-RN thermodynamics and instabilities, see [64–66]. For simplicity, we work in asymptotically flat spacetime in 4 dimensions. Since we do not want to treat the radiation in detail, we go to the regime where the mass emitted is large compared to the temperature. For convenience, we remain in the regime where the change in mass is small:

$$T \ll \Delta M \ll M\,. \tag{138}$$

In this regime, we expect that the rate is given by a Boltzmann-type factor. Computing the prefactor correctly would require a more careful treatment of the radiation but in the regime $\Delta M \gg T$ we can extract the Boltzmann factor.

For convenience, we describe the radiation as a thin shell. Domain walls that separate two different phases naturally have a brane equation of state $P = -\rho$, corresponding to a stress tensor simply proportional to the induced metric. On the other hand, a thin shell that models radiation should have its energy density decrease as it expands. A simple and natural choice for the equation of state is $P = \frac{1}{2}\rho$, corresponding to radiation living on the brane.

As an aside, one can begin with *free* massless particles living in the bulk spacetime and try to take a thin-wall limit. We have encountered obstacles in trying to take this limit- it appears that the thin wall limit is only consistent when the wall moves at the speed of light.[9] While this

---

[9]We thank Diego Hofman for discussions on this topic.

is sensible for massless particles, it is not conducive to a computation of Hawking radiation via tunnelling, which requires that the system have classically allowed and forbidden regions, and in particular turning points. One can think of our thin wall as modelling some massless particles that interact in such a way as to stay in equilibrium in a relatively thin shell. At a large radius, as the shell expands, the energy redshifts just like radiation. Since the exponential part of the tunnelling rate should not depend on the details of the particles being emitted, we are free to choose this convenient thin shell model, which makes backreaction computations tractable.

The tunnelling rate can be computed in a number of ways, which we detail below in order to establish the equivalence (and lack thereof) between them.

### 4.2.1 Probe calculation

Since $\Delta M / M$ is small, the radiation is a small perturbation on the background, and we can simply treat the shell as living in a fixed background. The action for the shell is

$$S = \int \rho r^2 d^2 \Omega_2 d\tau \,, \tag{139}$$

where $\tau$ is the proper time along the wall and $\rho$ is the energy density.

For spherically symmetric walls, the action is just

$$S = 4\pi \int \rho \, r^2 d\tau \,. \tag{140}$$

For our radiation wall, $\rho$ redshifts as the wall expands as $r^{-3}$, so

$$\rho = \frac{\sigma}{r^3} \,, \tag{141}$$

and the action is just

$$S = 4\pi\sigma \int \frac{d\tau}{r} \,. \tag{142}$$

It looks as though the equation of motion will be trivial, but note that the function we should vary is $r(t)$ rather than $r(\tau)$. They are related by

$$d\tau^2 = f dt^2 - \frac{dr^2}{f} \,. \tag{143}$$

To get the equation of motion it is convenient to think of $t$ rather than $r$ as the dynamical variable, so we use

$$d\tau = dr \sqrt{f t'^2 - 1/f} \,, \tag{144}$$

so that

$$S = 4\pi\sigma \int \frac{dr}{r} \sqrt{f t'^2 - 1/f} \,. \tag{145}$$

The equation of motion is then

$$\frac{4\pi\sigma f t'}{r\sqrt{f t'^2 - 1/f}} = E \,. \tag{146}$$

Having varied the action, we now go back to the more intuitive parameterization of the path in terms of $r(\tau)$, so that the equation is just

$$E = \frac{4\pi\sigma f}{r} \dot{t} = \frac{4\pi\sigma}{r} \sqrt{f + \dot{r}^2} \,. \tag{147}$$

This has an intuitive explanation as the energy of the shell; recalling the definition of $\sigma$, it is just

$$E = 4\pi r^2 \rho \sqrt{f + \dot{r}^2} \,, \tag{148}$$

where the square root can be thought of as a curved space gamma factor. Once we backreact this solution, we will see that this conserved quantity $E$ is indeed the energy associated with the shell, or equivalently the change in black hole mass.

The equation of motion of the shell is

$$\dot{r}^2 + f - \left(\frac{Er}{4\pi\sigma}\right)^2 = 0 \,. \tag{149}$$

This can be thought of as the equation of motion of a particle moving in a potential $V = f - \left(\frac{Er}{4\pi\sigma}\right)^2$. For many choices of the energy of the shell, there is a classically allowed and forbidden region of the potential, so we are in a good position to compute tunnelling. This equation describes the motion of the shell in Lorentzian space.

Assuming we choose an energy such that there is a forbidden region. The Euclidean equation of motion just differs by a sign,

$$\dot{r}^2 - f + \left(\frac{Er}{4\pi\sigma}\right)^2 = 0 \quad \text{(Euclidean)}. \tag{150}$$

**Static solution**

The Euclidean geometry is periodic in time, so our shell has to respect that periodicity. The simplest solution is simply to have the shell sit at the maximum of the Lorentzian potential. This means that we have to choose the conserved quantity $E$ so that the maximum of the potential has $V = 0$. We discuss whether this simplest solution is actually the dominant instanton in appendix F.

Let's call the location of the static solution $\hat{r}$. We need

$$f(\hat{r}) = \left(\frac{E\hat{r}}{4\pi\sigma}\right)^2 \,, \quad \text{and} \tag{151}$$

$$f'(\hat{r}) = 2\left(\frac{E}{4\pi\sigma}\right)^2 \hat{r} \,. \tag{152}$$

Combining these equations gives an equation for $\hat{r}$,

$$f'(\hat{r}) = 2\frac{f(\hat{r})}{\hat{r}} \,. \tag{153}$$

For Schwarzschild black holes, this simply corresponds to $r = 3GM$, and more generally probably still corresponds to the location of the innermost circular orbit.

Let's compute the action! Since the shell is static, it is not too difficult:

$$S_{\text{Euc}} = \frac{4\pi\sigma\Delta\tau}{\hat{r}} = \frac{4\pi\sigma\sqrt{\hat{f}}\Delta t_E}{\hat{r}} = \frac{4\pi\sigma\sqrt{\hat{f}}\beta}{\hat{r}} \,, \tag{154}$$

where we have denoted $f(\hat{r}) = \hat{f}$ and used the period of Euclidean time $\beta$. Using equation (151), we can write this in terms of the energy $E$ of the shell as

$$S_{\text{Euc}} = \beta E \,. \tag{155}$$

So, all of these shell computations serve only to compute the usual Boltzmann factor! This, however, is encouraging that our computations are sensible.

One might wonder whether there should be a correction due to the entropy of the shell. The shell has energy density $\rho \sim T^3$ and entropy $S \sim r^2 T^2$ where $T$ is the temperature of the radiation, so $S \sim \rho^{2/3} r^2$. Using $\rho \sim \sigma/r^3$, we have

$$\text{Entropy of shell} \sim \sigma^{2/3}. \tag{156}$$

At least for Schwarzschild black holes, this is subleading: the Euclidean action is $S_{\text{Euc}} \sim \sigma$ and we need to be in the regime where $S_{\text{Euc}} \gg 1$ for the tunnelling analysis to apply.

**Fixed shell energy density**

Given an initial black hole, it has some probability to emit a shell with a range of energies $E$. What we have computed here is the exponential part of the transition rate to a fixed final energy. Conventional domain walls are analogous to having a fixed $c$, but since our walls model radiation, it is reasonable to treat $\sigma$ as an adjustable parameter whose value determines how much energy is in the wall. Our proposal is that to emit energy $E$ we choose the value of $\sigma$ that minimizes the action.

In the above calculation, we held $\sigma$ fixed while allowing the trajectory to vary. When the trajectory varies, the conserved quantity $E$ varies. So the new instanton will describe emission of a different amount of energy, and tunnelling to a different final state.

What may be more sensible is to fix $E$, since we want to compute the rate of transitioning by a fixed amount (i.e. to a fixed final state). Of course, one could later integrate over the possible final states, if that is the physical quantity of interest. Although $E$ is fixed, it is reasonable to allow $\sigma$ to vary. This corresponds to changing the proper energy density in the wall in order to extremize the action. (For traditional domain walls interpolating between two vacua, we typically treat the domain wall tension as fixed; however, we could think of this tension as the result of a minimization procedure where we find the minimum action field configuration that interpolates between the vacua.)

So, we have motivated fixing $E$ and varying the tension parameter $\sigma$ in order to extremize the action. We do not have a general argument, but the problem is tractable for small perturbations around the static solution. We confine attention to solutions of the equation of motion (since we are looking for a saddle), and ask how the action depends on $\sigma$ for fixed $E$. Plugging the equation of motion into the expansion of the action (see appendix F) gives

$$S = 4\pi\sigma \int \frac{d\tau}{r} = 4\pi\sigma \int \frac{dt}{\dot{t}r} = \frac{(4\pi\sigma)^2}{E} \int dt \frac{f}{r^2}. \tag{157}$$

Recall that the static solution sits at an extremum of the integrand. So small perturbations around the static solution do not change the integrand to first order and only change the prefactor. This indicates that $\sigma$ should take the minimum possible value consistent with the existence of a Euclidean solution. Looking back at the Euclidean equation of motion

$$\dot{r}^2 - f + \left(\frac{Er}{4\pi\sigma}\right)^2 = 0, \tag{158}$$

it is evident that decreasing $\sigma$ increases the effective potential so that for $\sigma$ below some critical value there is no real Euclidean solution. The minimum allowed $\sigma$ gives precisely the static solution. Note that in the above analysis, we do not worry about whether the solution has the correct periodicity; this would further reduce the solutions under consideration.

So with this logic (fixed $E$), it appears that the static solution is an extremum. It would be nice to justify this further, for example via the gravity calculation.

### 4.2.2 Gravity calculation

We would now like to test our gravity formulas by comparing them to the above computation in the overlapping regime of validity. The equation of motion for the wall is given by the Israel junction condition,

$$\sqrt{f_i + \dot{r}^2} - \sqrt{f_o + \dot{r}^2} = 4\pi G\rho r = \frac{4\pi G\sigma}{r^2}, \tag{159}$$

where $f_i$ and $f_o$ denote the emblackening factors inside and outside the shell.

Let's again look for static shells. These satisfy

$$\sqrt{f_i(\hat{r})} - \sqrt{f_o(\hat{r})} = \frac{4\pi G\sigma}{\hat{r}^2}, \quad \text{and} \tag{160}$$

$$\frac{d}{dr}\left(\sqrt{f}_i - \sqrt{f}_o\right)\Big|_{\hat{r}} = -2\frac{4\pi G\sigma}{\hat{r}^3}. \tag{161}$$

Since we already used the approximation $\Delta M \ll M$ above, we use it here as well to simplify, although it is not needed in principle. The two metrics differ only by their mass, so let

$$f_i = f + \Delta f, \quad f_o = f - \Delta f, \quad \text{and} \quad \Delta f = G\Delta M/r, \tag{162}$$

with $\Delta M = M_o - M_i$. The equations become

$$\frac{G\Delta M}{\hat{r}\sqrt{\hat{f}}} = \frac{4\pi G\sigma}{\hat{r}^2}, \quad \text{and} \tag{163}$$

$$f'(\hat{r}) = 2f(\hat{r})/\hat{r}. \tag{164}$$

The second condition is the same condition for $\hat{r}$ as before, while the upper condition confirms that the change in mass appearing in the metric agrees with the conserved quantity $E$ of the probe computation,

$$\Delta M = 4\pi\sqrt{\hat{f}}\sigma/\hat{r} = 4\pi r^2\rho\sqrt{\hat{f}} = E. \tag{165}$$

Now we should compute the action!

It is at this point that we open the can of worms regarding the boundary conditions in gravity.

**Boundary conditions**

The most common boundary conditions (which seem to be the most sensible in asymptotically AdS spacetime) involve fixing the metric at infinity. This requires adding the GHY boundary term, as reviewed above.

Even before computing the action, we encounter an issue with our RN black holes. Since we are fixing the metric at infinity, we fix the temperature $\beta$. We then do not get to fix the mass at infinity. Our static instanton has interior mass $M_i$ and exterior mass $M_o$. The period of Euclidean time in the interior region is given by demanding regularity at the tip of the cigar, $\beta_i = 4\pi/f'(r_+)$. The exterior period is then determined by matching the geometry at the location of the domain wall. The proper imaginary time of the domain wall must agree inside and outside, requiring

$$\sqrt{\hat{f}_i}\beta_i = \sqrt{\hat{f}_o}\beta_o. \tag{166}$$

This equation determines $\beta_o$.

For Schwarzschild black holes, a lucky accident occurs and the $\beta_o$ determined from this equation happens to agree with the mass for a black hole with inverse temperature $\beta_o$. However, for RN black holes, these do not agree.

So for RN black holes, our instanton has a geometry near infinity which corresponds to a black hole with mass $M_o$ but with the 'wrong' periodicity in imaginary time, $\beta_o \neq \beta(M_0)$. There is nothing illegal about this: within the canonical ensemble, this instanton computes the rate for a process where the temperature at infinity is held fixed at $\beta_0$. (In all these computations, the charge is held fixed, and the domain walls are uncharged.) The relevant background solution is a black hole with period $\beta_0$ and the corresponding mass, $\tilde{M} \equiv M(\beta_0)$ determined by the RN solution.

So this instanton computes the rate of a process where a black hole with initial mass $\tilde{M}$ exchanges some energy with the heat bath, and nucleates a shell of radiation with energy $\Delta M$, in a geometry with interior mass $M_i$ and exterior mass $M_o$. We can compute the Euclidean action for this process. We now need to include the gravitational and electromagnetic contributions to the action.

Several simplifications are possible in computing the gravity action, which all arise from the statement that the Hamiltonian is a boundary term in gravity. Hawking and Horowitz [67] showed that the integral of the gravity action over a region simplifies as

$$\int_M \sqrt{g}R = \int_M \sqrt{g}(^3R - K^2 + K_{ab}K^{ab}) + 2\int_{\partial M} \sqrt{h}n_a u^b \nabla_b u^a. \tag{167}$$

Here, we have decomposed the manifold in a $3+1$ decomposition, $n^a$ is the outward pointing unit normal to the boundary, and $u^a$ is the unit normal to the time slices. An important point is that since we have used the $3+1$ decomposition, we need to include locations where this slicing degenerates (i.e. the horizon) as part of $\partial M$, even though from a covariant point of view there is no boundary at this location, and no localized term in the action.

The nice simplification is that for static solutions, the bulk terms cancel with the sources due to the Hamiltonian constraint. Gregory, Moss, and Withers [68] gave a convenient formula for geometries of the form

$$ds^2 = -f(r)dt^2 + \frac{dr^2}{f(r)} + r^2 d\Omega_2^2. \tag{168}$$

The Euclidean gravity action becomes

$$-\frac{1}{16\pi G}\int_M \sqrt{g}R = \text{bulk} + \frac{1}{16\pi G}\int_{\partial M} \sqrt{h}n_r f'(r). \tag{169}$$

This formula applies even if we have a static region of spacetime bounded by a non-static domain wall. Again, $n^a$ is the outward pointing unit normal to the boundary, and $h$ is the determinant of the induced metric on the boundary.

Since the bulk term in the action will cancel with the matter Lagrangian, the full action will be given by just the boundary terms. It is convenient to compute this in special cases. Evaluated on the black hole horizon, we get

$$S_{\text{hor}} = -\frac{A}{4G}. \tag{170}$$

Evaluated at infinity, we get (for asymptotically flat black holes)

$$S_\infty = \frac{1}{2}\beta M, \tag{171}$$

where $\beta$ is the period of Euclidean time, and $M$ is the asymptotic mass of the solution.

With this simplification for computing the bulk action in hand, we can compute the full action for the static domain walls.

For general moving shells, we will need to decompose the action into an integral over the interior and exterior regions in order to use the Gregory et al formula. But for static walls, we can simply include the thin shell as part of the matter action so that

$$S_{\text{Euc}}^{\text{bulk}} = \int \sqrt{g}\left[-\frac{R}{16\pi G} + \mathcal{L}_m\right] = -\frac{1}{8\pi G}\int_{\partial M} \sqrt{h}\, n_a u^b \nabla_b u^a$$
$$= \frac{1}{16\pi G}\int_{\partial M} \sqrt{h}\, n_r f'(r),$$

(172)

for static solutions. This can be further simplified to

$$S_{\text{Euc}}^{\text{bulk}} = -\frac{A}{4G} + \frac{1}{2}\beta M_\infty.$$

(173)

**Microcanonical ensemble**

As described above, we can follow Brown and York in fixing the mass of the solution, as well as the size of the spatial sphere at infinity. This requires adding the boundary term

$$S_{\text{Euc,m}}^{\partial} = -\int_{\partial M} \sqrt{h}\,\frac{t_\mu K^{\mu\nu}\partial_\nu t}{8\pi G}.$$

(174)

Evaluating this term simply cancels the term at infinity coming from the evaluation of the bulk action [61,62] yielding simply

$$S_{\text{Euc}} = -\frac{A}{4G}.$$

(175)

To compute the rate, we need to subtract the background action from the action of the instanton. Our instanton has interior mass $M_i$ and exterior mass $M_o = M_i + \Delta M$. In the microcanonical ensemble, the mass at infinity is fixed, so we do not care too much about the exterior period of Euclidean time.

Since the mass is fixed, the background solution is simply the black hole with mass $M_o$. The action difference is therefore

$$\Delta S_{\text{Euc}} = \frac{A(M_o) - A(M_i)}{4G} = S_o - S_i,$$

(176)

where the last equation contains the entropies $S_o, S_i$ rather than action. Black hole thermodynamics guarantees that $\Delta S = \beta \Delta M$ when the charge and volume are fixed. (Or we could compute the areas explicitly.) Therefore, our result here agrees with the probe computation.

Note that in principle the question of the number of negative modes may have a different answer once we allow gravity to be dynamical. Fully characterizing the fluctuations in Euclidean quantum gravity is beset with subtleties due to an apparently infinite number of negative modes, related to the conformal factor problem. See [69] for interesting recent work on this. We will not delve into this issue here, and content ourselves with finding the saddles.

**Canonical ensemble**

For the canonical ensemble, we fix the boundary metric. In the action, we have to add the standard GHY term given above. Recall that our instanton has interior and exterior masses $M_i, M_o$ and inverse temperatures $\beta_i, \beta_o$. Recall that, except for the Schwarzschild case, there is a mismatch between $\beta_o$ and $M_o$. The background solution has the same $\beta_o$, but a different mass $\tilde{M} = M(\beta_o)$.

The GHY term is divergent for each solution separately, but the divergence cancels. Evaluating the GHY term gives

$$S_{\text{Euc, GHY}} - S^o_{\text{Euc, GHY}} = -\frac{1}{8\pi G}\int (K - K_0) = \frac{1}{2}\beta M. \tag{177}$$

Combining this with the bulk action gives

$$\Delta S_{\text{Euc, can}} = \beta_\infty \Delta M_\infty - \frac{\Delta A}{4G}. \tag{178}$$

This has a very natural thermodynamic interpretation:

$$\Delta S_{\text{Euc, can}} = \beta \Delta F, \tag{179}$$

where $F$ is the free energy.

The action difference becomes

$$\Delta S_{\text{Euc}} = \beta_o(M_o - \tilde{M}) + S(\tilde{M}) - S(M_i). \tag{180}$$

As a reminder: $\beta_o$ is the inverse temperature, which is held fixed. $M_o$ is the asymptotic mass *after* tunnelling, $M_i$ is the mass of the black hole after tunnelling, while $\tilde{M}$ is the asymptotic mass before tunnelling, which is determined by $\beta_o$.

This equation is sensible in the sense that it is simply related to the difference in free energies, but it does not agree with the probe computation above due to the presence of three different masses in the equation.

So, apparently, the probe calculation corresponds to the microcanonical computation, which may be the most physically sensible one. This is the main computation we will rely upon in our analysis of the traversable wormhole.

## 5 Tunnelling to traversable wormholes

Now we will apply the recipe introduced in section 4 to the traversable wormhole solution presented in this work. We imagine ourselves to be in the black hole phase of the phase diagram (fig. 1) after which we turn on the nonlocal coupling and the true ground state is given by the traversable wormhole. We then want to compute the decay rate for tunnelling to the new ground state. In section 5.1 we do this in the canonical ensemble, and in section 5.2 we consider the microcanonical ensemble.

### 5.1 Canonical ensemble

We want to consider the nucleation rate of a traversable wormhole at a finite temperature. The wormhole geometry in its $AdS_2 \times S^2$ throat region, $\epsilon \ll (r - \bar{r})/\bar{r} \ll 1$ is identical, up to and including quadratic order in $\epsilon$ and $(r - \bar{r})/\bar{r}$, to a super-extremal AdS-RN geometry of the same asymptotic $U(1)$ charge and a mass shift $\Delta M < 0$ [9]. Thus, the wormhole's emblackening factor is identical to that of an extremal AdS RN black hole of the same charge,

$$f_{ext.}(r) = \frac{1}{\ell^2}\left(\frac{r - \bar{r}}{r}\right)^2(\ell^2 + r^2 + 2r\bar{r} + 3\bar{r}^2), \tag{181}$$

except for a shift in the mass term,

$$f_i(r) = f_{ext.}(r) - \frac{2G\Delta M}{r}. \tag{182}$$

The geometry we are considering the decay of is an AdS Reissner-Nordström black hole. Our bounce solution has an exterior geometry which is AdS-RN with an emblackening factor

$$f_o(r) = f_{ext.}(r) - \frac{2G\Delta M_o}{r} \,, \tag{183}$$

where $\Delta M_o := M_o - M_{ext.}$ is the asymptotic mass above extremality of the exterior geometry specified by $f_o(r)$.

Suppose that there is a spherically symmetric gravitational instanton that mediates the decay from a black hole to a traversable wormhole. If we assume that the thin wall approximation is valid, the instanton geometry is approximated by the (Euclidean) wormhole geometry glued inside the (Euclidean) extremal AdS-RN geometry. The trajectory of a domain wall separating the two geometries is determined by the Israel junction conditions [51]. The equation of motion of a thin, spherically symmetric domain wall separating a geometry with emblackening factor $f_o$ on the outside and $f_i$ on the inside, is $\dot{R}^2 + V_{eff}(R) = 0$, with potential

$$V_{eff}(r) = f_o(r) - \frac{(f_i(r) - f_o(r) - \kappa^2 r^2)^2}{4\kappa^2 r^2} \,, \tag{184}$$

which for us evaluates to

$$V_{eff}(r) = f_{ext.}(r) - \frac{G}{r}(\Delta M + \Delta M_o) - \frac{G^2(\Delta M - \Delta M_o)^2}{\kappa^2 r^4} - \frac{\kappa^2 r^2}{4} \,. \tag{185}$$

We know that the wormhole geometry is approximately super-extremal AdS-RN, i.e. has emblackening factor (182), but only for $(r - \bar{r})/\bar{r} \gg \epsilon$, so the domain wall effective potential (185) cannot be trusted for smaller values of $r$. We discuss domain wall trajectories in the throat region in appendix B.

We assume that the surface stress tensor of the domain wall is that of a perfect fluid. The dependence of the wall's energy density $\kappa$ as a function of the radial position depends on the fluid equation of state. In appendix D we analyse the trajectory of a general perfect fluid domain wall. It has a radius-dependent energy density

$$\kappa(r) = \sigma r^{-\alpha} \,, \tag{186}$$

where $\sigma$ is a constant, and $\alpha = 0$ for the vacuum, $\alpha = 2$ for dust, and $\alpha = 3$ for radiation. We discuss domain wall trajectories for vacuum equation of state in appendix C.

We need to check the orientations of the outward normals, to derive the necessary condition to be gluing the correct parts of the spacetimes together. The Lorentzian domain wall equation of motion only has real solutions when $V_{eff} < 0$. To find the turning points of the domain walls, we need to find the roots of the potential. Sign information is lost in deriving the domain wall effective potential, so, besides finding a solution for the wall trajectory in the effective potential, we also need to check the orientation of the outward normals of the geometries we are glueing together [51]. We want to confirm that we are glueing the interior $r < R$ of the wormhole geometry to the exterior $r > R$ of AdS-RN and not, say, interior to interior, so we need to check the direction of the outward normals by calculating the signs of the $\theta\theta$ component of the extrinsic curvatures, which is denoted by $\beta$ in the literature. For the inner geometry,

$$2\kappa R \beta_i = f_i(R) - f_o(R) + \kappa^2 R^2 = \frac{2G(-\Delta M + \Delta M_o)}{R} + \kappa^2 R^2 \,, \tag{187}$$

while for the exterior geometry

$$2\kappa R \beta_o = f_i(R) - f_o(R) - \kappa^2 R^2 = \frac{2G(-\Delta M + \Delta M_o)}{R} - \kappa^2 R^2 \,. \tag{188}$$

To have both $\beta_o$ and $\beta_i$ positive, and so have both outwards normal vectors point in the positive $r$ direction, requires

$$\frac{2G(-\Delta M + \Delta M_o)}{R} \pm \kappa^2 R^2 \geq 0, \tag{189}$$

for all $R$ in the range of the domain wall trajectory. This condition is valid in both Lorentzian and Euclidean signatures. Two immediate consequences of this condition are that the asymptotic mass cannot be lower than the wormhole mass, and that if the domain wall energy density is $R$-independent then the wall cannot reach the asymptotic boundary.

### 5.1.1 Static domain wall

For fixed asymptotic temperature $\beta$, non-static domain walls must have a Euclidean time periodicity $\beta/\mathbb{N}$, otherwise, their radial position is multi-valued as a function of Euclidean time. In contrast, any static domain wall is a consistent solution.

For a given domain wall energy density $\kappa$ and asymptotic temperature $\beta$, allowed asymptotic masses of the bounce solution $M_o$ are found by solving the Israel junction equation

$$\sqrt{f_i(R) + \dot{R}^2} - \sqrt{f_o(R) + \dot{R}^2} = \kappa R. \tag{190}$$

Finding the allowed values of $M_o$ for a given $\kappa$ and $\beta$ is not straightforward, especially with the constraint on the Euclidean time periodicity. The condition on the outward normals (189) for static walls is satisfied if and only if[10]

$$\Delta M_o \geq \Delta M. \tag{191}$$

If this is satisfied then the energy density of the wall $\kappa$ is also non-negative. If we assume that the domain wall is static, $\dot{R} = 0$, then we are at a double zero of the effective potential:

$$V_{eff}(R)|_{R=\hat{R}} = 0, \tag{192}$$
$$V'_{eff}(R)|_{R=\hat{R}} = 0, \tag{193}$$

where $\hat{R}$ is the radius that the static wall sits at. We discuss non-static domain walls in appendix E. Solving (192) is equivalent to solving (190) with $\dot{R} = 0$, which gives

$$\Delta M_o = \Delta M + \frac{\kappa^2 \hat{R}^3}{2G} \left( \frac{2}{\kappa \hat{R}} \sqrt{f_{ext.}(\hat{R}) - \frac{2G\Delta M}{\hat{R}}} - 1 \right). \tag{194}$$

Solving (193) depends on the $R$ dependence of $\kappa$; if we assume a radiation equation of state then the equation to solve is

$$\frac{d}{dR} \left( R^2 \left( \sqrt{f_i(R)} - \sqrt{f_o(R)} \right) \right)|_{R=\hat{R}} = 0. \tag{195}$$

We can solve the pair of equations to find the external mass as a function of the energy density of the static radiation-composed domain wall. This is a complicated formula in general, [11] but we can get a simple approximation when $\sigma$ is large:

$$G\Delta M_o \approx \frac{\sigma}{l}, \tag{196}$$

---

[10]To see this, note that (190) is $\sqrt{f_i} - \sqrt{f_o} = \kappa R$ for a static wall, and that (189) can be written as $f_i - f_o \pm \kappa^2 R^2 \geq 0$.

[11]We also calculated the discriminant of the effective potential as a function of $R$, which immediately gives a necessary condition for there to be a static wall, *i.e.* for the effective potential to have a double zero, but we also found the discriminant as a function of the interior and exterior masses to be too complicated to be useful.

and

$$\hat{R} \approx \frac{3}{2}\frac{\sigma}{l}\,. \tag{197}$$

This large $\sigma$ domain wall is a dilute shell of radiation gas with a large total mass but low energy density. Fortunately, this approximation will not be needed in the final computation of the action.

### 5.1.2 Nucleation rate

The nucleation rate is

$$\Gamma \sim e^{-B/\hbar}\,, \tag{198}$$

where the tunnelling exponent $B$ is the difference between the Euclidean actions of the bounce and false vacuum solutions:

$$B = S_{bounce} - S_{f.v.}\,. \tag{199}$$

Both actions include bulk and boundary terms. The bulk Lagrangian is

$$\mathcal{L}_{bulk} = -\frac{1}{16\pi G}(R - 2\Lambda) + \mathcal{L}_{matter}\,, \tag{200}$$

where our $\mathcal{L}_{matter}$ has Dirac fermions and a $U(1)$ gauge field. The appropriate boundary Lagrangian to use depends on which boundary conditions we impose. Since we are working with the canonical ensemble, we want to fix the asymptotic Euclidean time periodicity. If we fix the asymptotic temperature with Dirichlet boundary conditions on the boundary metric, then the boundary Lagrangian has a gravitational GHY term, boundary terms for the matter fields, and counterterms to cancel divergences in the action,

$$\mathcal{L}_{bdy} = -\frac{1}{8\pi G}K + \mathcal{L}_{matter,bdy} + \mathcal{L}_{c.t}\,. \tag{201}$$

Our sign convention has the normal to $\partial M$ pointing in the direction of increasing $r$. If were to fix the asymptotic mass, as relevant for the microcanonical ensemble, then the appropriate boundary term for the gravitational action would be (135).

Both the false and true vacuum solutions are static so we can foliate both $M_{tv}$ and $M_{fv}$ with time slices $\Sigma$ on which the time derivatives of the 3-metric and matter fields vanish. We can make use of this to simplify the evaluation of the on-shell action. The false vacuum solution is two copies of Euclidean AdS-RN, both with thermal periodicity $\beta$. To be a genuine on-shell solution, the mass of the black hole is fixed once $\beta$ is fixed. The false vacuum action is

$$S_{f.v.} = \int_{M_{fv}} \mathcal{L}_{bulk} + \int_{\partial M_{fv}} \mathcal{L}_{bdy}\,. \tag{202}$$

The Euclidean time slices of $M_{fv}$ intersect at the black hole horizon. Let us split the integral over $M_{fv}$ by excising an infinitesimal ball-shaped region centred on the horizon. The gravitational action of such an infinitesimal ball $B$ is [68,70]

$$-\frac{1}{16G}\int_B R - \frac{1}{8\pi G}\int_{\partial B} K = \frac{A_h}{4G}\,. \tag{203}$$

The complementary region $M_{fv} - B$ has an inner boundary at the black hole horizon and an outer boundary at asymptotic infinity. We can write the Ricci scalar on the slices in terms of the extrinsic and intrinsic 3-geometry of those slices using the contracted form of the Gauss-Codazzi equations

$$R = \mathcal{R}^{(3)} - K^2 + K_{ab}K^{ab} - 2\nabla_a(u^a\nabla_b u^b) + 2\nabla_b(u^a\nabla_a u^b)\,, \tag{204}$$

where $u$ is the unit normal to the slices. The $3+1$ form of the Lagrangian is

$$\mathcal{L}_{bulk} = \frac{1}{16\pi G}(\dot{g}_{ij}^{(3)}\pi_{ij} + \dot{\phi}\pi_\phi - N\mathcal{H} - N^i\mathcal{H}_i) + \frac{1}{8\pi G}\left(\nabla_a(u^a\nabla_b u^b) - \nabla_b(u^a\nabla_a u^b)\right). \quad (205)$$

Here $\phi$ represents the matter fields with $\pi_\phi$ their conjugate momenta, and dots are derivatives with respect to Euclidean time. The Hamiltonian constraint $\mathcal{H}$ and momentum constraint $\mathcal{H}_i$, which depend on the matter content, vanish on-shell. Because our solution is static, the terms with time derivatives also vanish.

There are two total derivative terms in (205). The first does not contribute: $u^a\nabla_b u^b$ is proportional to $u$ and cancels the GHY terms on the initial and final time slices. The second total derivative term is important: $u^a\nabla_a u^b$ is orthogonal to $u$ and combines with the GHY terms $K = \nabla_a r^a$ on the inner and outer boundaries, where $r$ is the unit normal to the boundaries of the time slice, to give a term proportional to the extrinsic curvature of the 2d boundaries of the time slices [67]

$$\int_0^\beta d\tau \int_{\partial\Sigma} (\nabla_a r^a + r_b u^a \nabla_a u^b) = \int_0^\beta d\tau \int_{\partial\Sigma} (\delta_b^a - u^a u_b)\nabla_a r^b$$
$$= \int_0^\beta d\tau \int_{\partial\Sigma} {}^{(2)}K. \quad (206)$$

The contribution of this term to the inner boundary of $M_{fv} - B$ is negligible [68]. Our false vacuum action has so far simplified to

$$S_{f.v.} = 2\left[-\frac{A_h}{4G} - \frac{1}{8\pi G}\int_{\partial M_{fv}} ({}^{(2)}K - \mathcal{L}_{c.t.})\right]. \quad (207)$$

The holographic counter term for AdS$_4$ is [27]

$$\mathcal{L}_{c.t.} = \frac{2}{\ell}\left(1 + \frac{\ell^2}{4}{}^{(3)}\mathcal{R}\right). \quad (208)$$

Recall that $M_{f.v.}$ is the geometry of a near-extremal AdS-RN black hole. The boundary terms on a $r = r_c$ radial cutoff surface evaluate to

$$\int {}^{(2)}K = 8\pi\beta\left(\frac{r_c^3}{\ell^2} + r_c - 2GM + O(r_c^{-1})\right), \quad (209)$$

and

$$\int \mathcal{L}_{c.t.} = 8\pi\beta\left(\frac{r_c^3}{\ell^2} + r_c - GM + O(r_c^{-1})\right), \quad (210)$$

where $M$ is the asymptotic mass. The divergences in the boundary terms cancel, as expected. The final result for the false vacuum action is

$$S_{f.v.} = 2\left[-\frac{A_h}{4G} + \beta M\right]. \quad (211)$$

This is the expected result: the total free energy of the two black holes.

Our bounce action has the same bulk and boundary Lagrangians as the false vacuum action. We evaluate these Lagrangians on the bounce solution:

$$S_{bounce} = \int_{M_{bounce}} \mathcal{L}_{bulk} + \int_{\partial M_{bounce}} \mathcal{L}_{bdy}. \quad (212)$$

The bounce solution $M_{bounce}$ has an interior and exterior geometry, glued together by domain walls. It has the traversable wormhole geometry of mass $M_{ext.} + \Delta M$ in the interior and two AdS-RN black holes of mass $M_o$ in the exterior

$$M_{bounce} = \text{Int}(M_{WH}) \cup \text{Ext}(M_{RN}). \tag{213}$$

Here $M_{bounce}$ is the traversable wormhole solution for $0 \leq r < R(\tau)$ and AdS-RN for $r > R(\tau)$, where $r = R(\tau)$ is the (in general time-dependent) radial position of the domain wall.

For a bubble solution with a static domain wall, the bulk Lagrangian reduces to the total derivative terms in (205), just as it did for the black hole. The action of the bubble is identical to (207) except that: (1) unlike the black hole, the bounce solution has the horizonless traversable wormhole geometry in the interior, so there is no horizon contribution to the on-shell action, and (2) the asymptotic mass is $M_o$, not $M$, the mass of an AdS-RN black hole of inverse temperature $\beta$. The resulting action is

$$S_{bounce} = 2\beta M_o, \tag{214}$$

and the tunnelling exponent is

$$\boxed{B = 2\beta(M_o - M) + 2\frac{A_h}{4G}.} \tag{215}$$

This is the main result of this subsection. It is the exponent for the finite temperature transition rate from a pair of AdS-RN black holes with masses $M(\beta)$ to the traversable wormhole geometry glued, by a static domain wall, inside of AdS-RN with asymptotic mass $M_o$. Since we are at a fixed temperature, not fixed energy, $M_o$ is free; however, our decay rate shows that fluctuations to higher energies, $M_o > M$, are Boltzmann-suppressed. The transition destroys the black hole, and this leads to the large entropic suppression given by the horizon area in the exponent.

The low temperature limit of (215) needs a comment because, for low values of the asymptotic mass $M_o$ in the bounce solution, the transition rate apparently diverges. The minimum allowed value of $M_o$ saturates the bound (191) and is given by the wormhole mass $M_{ext.} + \Delta M$, and the low temperature limit of the rate exponent for this decay channel diverges like $B \sim 2\beta \Delta M$. However, (215) is not valid outside of the regime of validity of the semiclassical approximation within which we calculated our decay exponent. This semiclassical breakdown temperature was given in (74), and our decay exponents are positive above this temperature, giving reasonable and small decay rates, even for the minimum allowed $M_o$. This is one resolution. It would be interesting to extend the decay rate exponent to low temperatures outside of the semiclassical regime, for example along the lines of [71].

## 5.2 Microcanonical ensemble

We now want to consider the tunnelling from the zero temperature extremal AdS RN black hole to a traversable wormhole of the same charge, at fixed asymptotic mass. For this, we need to be in the region of the $(h, \mu)$ phase diagram where the extremal black hole is a false vacuum and the traversable wormhole is the true vacuum; $h > 0$ and $\mu > \mu_c$ suffices. With mild assumptions, dynamical topology change in semiclassical gravity is forbidden [1–3]. To evolve from a pair of disconnected extremal black holes to a traversable wormhole requires a change in topology, so the dominant decay channel will be quantum mechanical; tunnelling via gravitational instanton. We want to find the tunnelling rate and the trajectory of the domain wall after the tunnelling event.

The geometry we are considering the decay of is an AdS Reissner-Nordström black hole with an emblackening factor

$$f_o(r) = 1 + \frac{r^2}{\ell^2} + \frac{r_e^2}{r^2} - \frac{2GM}{r}. \tag{216}$$

Our bounce solution, whose action determines the decay rate, is our traversable wormhole geometry in the interior and extremal AdS-RN with metric (216) in the exterior. The wormhole's emblackening factor is identical to that of the extremal black hole (216), except for a shift in the mass term,

$$f_i(r) = f_o(r) - \frac{2G\Delta M}{r}. \tag{217}$$

### 5.2.1  Domain wall effective potential

The trajectory of a domain wall separating the two geometries is determined by the Israel junction conditions [51]. The equation of motion of a thin, spherically symmetric domain wall separating a geometry with emblackening factor $f_0$ on the outside and $f_i$ on the inside, is $\dot{R}^2 + V_{eff}(R) = 0$, with potential

$$V_{eff}(r) = f_o(r) - \frac{(f_i(r) - f_o(r) - \kappa^2 r^2)^2}{4\kappa^2 r^2} \tag{218}$$

$$= \frac{2G\Delta M}{r} + \frac{f_o + f_i}{2} - \frac{\beta_o^2 + \beta_i^2}{2}, \tag{219}$$

which for us evaluates to

$$V_{eff}(r) = \left(\frac{1}{\ell^2} - \frac{\kappa^2}{4}\right)r^2 + 1 - \frac{2GM + G\Delta M}{r} + \frac{r_e^2}{r^2} - \frac{G^2\Delta M^2}{\kappa^2 r^4}. \tag{220}$$

Making use of the extremality of the exterior geometry, we can write the effective potential in terms of the extremal radius and $\epsilon$, which quantifies the proximity to extremality of the super-extremal AdS-RN approximation of the wormhole geometry:

$$V_{eff}(r) = \frac{1}{\ell^2}\left(\frac{r - \bar{r}}{r}\right)^2(\ell^2 + r^2 + 2r\bar{r} + 3\bar{r}^2) - \frac{1}{4}\left(\kappa r - \frac{\bar{r}\epsilon^2 \mathcal{C}(\bar{r})}{\kappa r^2}\right)^2. \tag{221}$$

To trust the approximations made in deriving this effective potential, we require $(r - \bar{r})/\bar{r} \gg \epsilon$, since otherwise (182) is not expected to be approximately true. We generalize this assumption in appendix B.

Now we derive the necessary condition in order to have the correct outward normal orientation. The $\theta\theta$ component of the extrinsic curvatures of the interior and exterior geometries are

$$2\kappa R\beta_i = f_i(R) - f_o(R) + \kappa^2 R^2 = -\frac{2G\Delta M}{R} + \kappa^2 R^2, \tag{222}$$

and

$$2\kappa R\beta_o = f_i(R) - f_o(R) - \kappa^2 R^2 = -\frac{2G\Delta M}{R} - \kappa^2 R^2. \tag{223}$$

To have both $\beta_o$ and $\beta_i$ positive, and so have both outwards normal vectors point in the positive $r$ direction, requires

$$\frac{2G(-\Delta M)}{R} \pm \kappa^2 R^2 \geq 0, \tag{224}$$

for all $R$ in the range of the domain wall trajectory. This condition is valid in both Lorentzian and Euclidean signatures.

### 5.2.2  Tunnelling rate

The instanton tunnelling rate is

$$\Gamma \sim e^{-B/\hbar}, \tag{225}$$

where

$$B = S_{bounce} - S_{f.v.}, \tag{226}$$

is the difference between the Euclidean actions of the bounce and false vacuum solutions.

The false vacuum solution is two copies of Euclidean AdS-RN. The false vacuum action is

$$S_{f.v.} = \int_{M_{fv}} \mathcal{L}_{bulk} + \int_{\partial M_{fv}} \mathcal{L}_{bdy}. \tag{227}$$

Since we fix the asymptotic mass then the appropriate boundary term for the gravitational action is (135).

Both the false and true vacuum solutions are static so we can foliate both $M_{tv}$ and $M_{fv}$ with time slices on which the time derivatives of the 3-metric and matter fields vanish. We can make use of this to simplify the evaluation of the on-shell action. The final result for the false vacuum action only has contributions from the horizon

$$S_{f.v.} = 2\left(-\frac{A_h}{4G}\right). \tag{228}$$

The instanton solution $M[\Sigma]$ has two domain walls glueing together two copies of AdS-RN with disks cut from their centres and our Euclidean traversable wormhole solution,

$$M[\Sigma] = \mathrm{Int}_\Sigma(M_{tv}) \cup \mathrm{Ext}_\Sigma(M_{fv}), \tag{229}$$

which is the traversable wormhole solution for $0 \le r < R(\tau)$ and AdS-RN for $r > R(\tau)$. We are working at fixed charge and temperature which, for regularity, fixes the asymptotic mass of the AdS-RN geometry $M_{fv}$. When glueing Euclidean black hole solutions together, one has to check that the instanton boundary conditions are consistent with a smooth geometry. For us, the size of the transverse sphere in the wormhole geometry never becomes zero, so there is no constraint on its thermal periodicity from regularity, and the instanton solution can have the same asymptotic boundary conditions, i.e. temperature and mass, as $M_{fv}$.

Now we can calculate the on-shell action of the bubble solution. Two key differences compared to the calculation of the false vacuum action are that there is a domain wall and that there is no inner horizon. The action has bulk, boundary, and wall contributions,

$$S_{bounce} = \int_{M[\Sigma]} \mathcal{L}_{bulk} + \int_{\partial M[\Sigma]} \mathcal{L}_{bdy} + \int_\Sigma \mathcal{L}_{wall}. \tag{230}$$

We can split the integral over $M[\Sigma]$ into integrals over the regions inside and out of the domain wall. In general, doing so introduces a difference of GHY terms localised to the wall coming from the gravitational action, but these vanish for us. This is because the wall is composed of radiation; taking the trace of the junction condition gives $\Delta K \propto \mathrm{Tr}(S)$, and the surface stress tensor for radiation is traceless in four dimensions, so $\Delta K = 0$.[12] We can also just integrate the bulk action over the wall directly. While the bulk stress tensor has a delta function singularity

---

[12]Since perfect fluid stress tensors with a radiation equation of state are only traceless in 4d, and the wall is 3d not 4d, the claim that our wall surface stress tensor is traceless may be surprising. Nonetheless, from the definition in appendix D, one can show that $\mathrm{Tr}(S) = 0$, in part because the normal components of the surface stress tensor are zero.

at the wall, which would usually give a finite contribution to the gravitational action, for us that tensor is traceless so the Ricci scalar in the gravitational Lagrangian on the wall vanishes.

We have assumed that the surface stress tensor of the bubble wall is a perfect fluid with a radiation equation of state.[13] The Lagrangian of a perfect fluid is proportional to its energy density [72, 73]. For a radiation equation of state and our conventions, this gives

$$\mathcal{L}_{wall} = \frac{1}{4\pi G} \frac{\sigma}{R^3}\,.\tag{231}$$

The decay exponent for a general radiation-composed domain wall is thus

$$\frac{B}{2} = \frac{A_h}{4G} + \frac{1}{4\pi G} \int_{\Sigma} \frac{\sigma}{R^3} - \frac{1}{16\pi G} \int_{\Sigma} \left( \frac{df_2(R)}{dR}\frac{d\tau_2}{d\tau} - \frac{df_1(R)}{dR}\frac{d\tau_1}{d\tau} \right)\,.\tag{232}$$

### 5.2.3 Static bubble solution and tunnelling rate

The domain wall potential has a double zero when $\sigma = \sigma_{static}$ where

$$\frac{\sigma_{static}}{\sigma_c} = \frac{4\bar{r}(\ell^2 + 3\bar{r}^2)^{3/2}}{(\ell^2 + 2\bar{r}^2)^{1/2}(\ell^2 + 6\bar{r}^2)^{3/2}}\,.\tag{233}$$

For this value of the energy density, the potential has a static domain wall solution that sits at radius

$$R_{static} = 2\bar{r} + \frac{6\bar{r}^3}{\ell^2}\,.\tag{234}$$

For the static domain wall, the integrands in (232) cancel and the decay exponent is simply

$$\boxed{\frac{B}{2} = \frac{A_h}{4G}}\,.\tag{235}$$

To see the cancellation, note that the time dilation factors for the static domain wall are $\dot{\tau}_{1,2} = (f_{1,2}(R))^{-1/2}$, and that

$$\left( \frac{f_2'}{\sqrt{f_2}} - \frac{f_1'}{\sqrt{f_1}} \right)\bigg|_{R=R_{static}} = 4\frac{\sigma_{static}}{R_{static}^3} + O(\epsilon^4)\,.\tag{236}$$

This cancels against the wall action term in (232) to give the decay exponent (235).

The result (235) can be written as $\Gamma \sim e^{-\Delta S}$ where $\Delta S$ is the total entropy of the black hole pair we are tunnelling from. Even though the wormhole has lower energy than the pair of black holes, tunnelling to it is entropically suppressed.

## 6 Discussion

In this paper, we have studied some nonperturbative effects of quantum gravity in a system of a coupled pair of holographic CFTs. There are three relevant phases: thermal AdS, a pair of extremal RN-AdS black holes and a traversable wormhole. We study the phase diagram of the system in detail, and in particular, we compute the transition rate for the transition of the black hole phase to the wormhole phase, which constitutes a nonperturbative effect of quantum gravity due to the difference in topology between the states. A very large number of open questions and future directions remain. We conclude by describing some of them:

---

[13]It would be interesting to generalize beyond this assumption. We discuss some basic results to this effect in appendices C and D.

**Simulating black holes with matrix models**  Recently, [74] estimated the number of qubits needed to simulate black hole features in the lab/on a quantum computer using a matrix model. It would be interesting to understand whether you would need more qubits to see the transitions discussed in this work. The upshot of keeping the degrees of freedom as small as possible, while still simulating gravitational physics is that effects that are (exponentially) suppressed in $G$ are not as suppressed as in the real world, where $G$ is very small.

Our system provides a valuable complementary approach to the wormholes constructed using the Gao-Jafferis-Wall protocol [11], adapted to the 2d setup of Maldacena and Qi [12] in [75, 76]. Recent claims to have simulated these wormholes [77] have been controversial, in part due to the difficulty in preparing the initial state [78]. Our setup provides a simple way to prepare the wormhole state, as the ground state of a simple Hamiltonian. A tradeoff is that our dual theory is higher dimensional, which is harder to simulate. It would be interesting to combine these approaches to try to have the best of both worlds: a lower dimensional system where the traversable wormhole is still the ground state of a simple Hamiltonian.

**Wormholes in the lab**  One of the main upshots of our work is that it provides a computation that seems possible to test experimentally. In order to do so one just needs to simulate/construct two copies of a holographic CFT, couple them, and implement time evolution according to our proposed Hamiltonian. We imagine dialling the chemical potential such that the ground state is a pair of extremal black holes. Then we turn on the non-local coupling and the system presumably transitions dynamically to the traversable wormhole, with the transition rate as computed in section 5.

**Instabilities/relevant deformations of AdS$_2$**  In many constructions there are relevant operators present that deform the infrared away from pure AdS$_2$; see e.g. [79]. When these operators are present, they will modify the description we have presented here. It would be interesting to analyze the effects of these operators.

**Fragmentation**  At the level of our analysis in this paper, wormholes with sufficiently small charges are energetically allowed to fragment into smaller wormholes. Naively, this process would seem to increase the entropy and therefore be the preferred ground state. A more careful computation is needed in this regime.

**Negative modes of the instanton**  Our instanton is static. Especially when the temperature becomes low, one might expect that there is an instability towards developing oscillations in Euclidean time. These would appear as extra negative modes of the instanton.

**Gravitational instanton prescription**  Additionally, there are some technical and conceptual questions related to our instanton computations. In the canonical ensemble calculation, it is not clear to us what fixes the external mass of the instanton, since the usual boundary conditions fix the temperature rather than the mass. In order to get a sensible answer, we assume the asymptotic mass of the instanton should match the initial asymptotic mass of the black hole. While this seems quite sensible physically, it is not clear to us how the prescription for computing decay rates enforces this choice. This ambiguity does not arise for tunnelling between different black holes, because regularity at the horizon fixes the additional ambiguity, whereas the traversable wormhole geometry does not have any preferred periodicity in Euclidean time.

**Quantum effects**  A crucial question is whether our semi-classical analysis here gives the correct decay rate. As we described, the temperature below which the wormhole dominates

the canonical ensemble is so low that the semi-classical description of the black hole breaks down above the transition temperature. It would be interesting to incorporate quantum effects in our analysis along the lines of [71, 80–82]

**Supersymmetry** Embedding our system in a supersymmetric theory would add additional theoretical control. It would be interesting to see whether this can be done in a simple manner and if and how it would affect the computations in this work. Additional charges would need to be added, and BPS black holes in asymptotically AdS spacetime must rotate (see e.g. [83]). At this point, it is not altogether clear whether supersymmetry is important in the regimes we study. One reason to suspect that it does play an important role is the disparity between the density of states near extremality between SUSY and non-SUSY black holes [80–82].

**Graviton mass** Since our model is constructed by coupling two different theories, one might worry that the coupling induces a graviton mass. This occurs for example in situations where entanglement wedges contain islands, which appear in AdS theories coupled to a bath [84]. We believe that in our model the graviton remains massless, because the coupled CFTs form a closed system, and there is no way for energy to leave it. It would be interesting to analyze this question more carefully.[14]

**Entanglement entropy** We can use the Ryu-Takayanagi (RT) prescription to calculate the entanglement entropy of boundary subregions and determine the properties of the entanglement structure of the different ground states [85]. In the black hole thermofield double state, the result is standard: the RT surface is the black hole horizon and the entanglement entropy $S_{L,R}$ of the state reduced to either of the CFTs equals the black hole entropy. In the wormhole state, there are no horizons and the RT surface is the $S^2$ sphere in the middle of the wormhole throat. From (23), this gives an entanglement entropy $S_{L,R}$ that is greater than the black hole entropy by the factor $(1+\zeta)$. The increased entanglement is related to the increased connectedness of the spacetime.

**CFT description of the wormhole** Although the wormhole is the ground state of a simple Hamiltonian, we do not have an explicit description of the CFT state. Presumably, the state looks something like a thermofield double state with chemical potential, but it must differ slightly because the thermofield double is dual to the eternal black hole. The entanglement entropy computations mentioned above give one clue to the CFT state.

# Acknowledgments

We thank Luis Apolo, Alejandra Castro, Ping Gao, Jesse Held, Diego Hofman, Alexey Milekhin, Dominik Neuenfeld, Jan Pieter van der Schaar, Masataka Watanabe, and Manus Visser for useful discussions. SB thanks the Department of Applied Mathematics and Theoretical Physics (DAMTP), at the University of Cambridge for hospitality.

**Funding information** The work of SB is supported by the Delta ITP consortium, a program of the Netherlands Organisation for Scientific Research (NWO) that is funded by the Dutch Ministry of Education, Culture and Science (OCW), and in part by the National Science Foundation under Grant No. NSF PHY-1748958, the Heising-Simons Foundation, and the Simons Foundation (216179, LB). The work of BF is supported by ERC Consolidator Grant QUANTIV-IOL and by the Heising-Simons Foundation "Observational Signatures of Quantum Gravity" collaboration grant 2021-2817. The work of AR is supported by the Stichting Nederlandse Wetenschappelijk Onderzoek Instituten (NWO-I) through the Scanning New Horizons project.

---

[14]We thank Hao Geng for bringing this question to our attention.

## A Conventions

In the metric Ansatz (6) we can pick the following vielbein

$$e^1 = e^\sigma dt\,, \quad e^2 = e^\sigma dx\,, \quad e^3 = Rd\theta\,, \quad e^4 = R\sin\theta d\phi\,. \tag{A.1}$$

The spin connection $\omega^{ab}$ can then be found by solving

$$de^a + \omega^{ab} \wedge e^b = 0\,, \quad \omega^{ab} = -\omega^{ba}\,, \tag{A.2}$$

which has a solution given by the nonzero components

$$\omega^{12} = \sigma' dt\,, \quad \omega^{32} = R'e^{-\sigma}d\theta\,, \quad \omega^{42} = R'\sin\theta e^{-\sigma}d\phi\,, \quad \omega^{43} = \cos\theta d\phi\,. \tag{A.3}$$

Here a prime denotes differentiation with respect to the radial coordinate $x$. Moreover, we write spinors as tensor products between the spherical components and the time and radial components. This can be done with the convention for gamma matrices given below

$$\gamma^1 = i\sigma_x \otimes 1\,, \quad \gamma^2 = \sigma_y \otimes 1\,, \quad \gamma^3 = \sigma_z \otimes \sigma_x\,, \quad \gamma^4 = \sigma_z \otimes \sigma_y\,. \tag{A.4}$$

## B Domain wall trajectories in static, spherically symmetric space-times

We want to study the existence and dynamics of domain walls deep in the throat region of the wormhole geometry, $(r - \bar{r})/\bar{r} \lesssim \epsilon$, where the geometry is no longer approximately (super-extremal) AdS-RN. The traversable wormhole geometry is static and spherically symmetric, so can be written in the form

$$ds^2 = -f_1(r)dt^2 + \frac{dr^2}{f_2(r)} + r^2 d\Omega_2^2\,. \tag{B.1}$$

The analysis of domain wall dynamics in [51] applies to geometries which can be written in the form where $f_1(r) = f_2(r)$, but, unlike AdS-RN and other spherically symmetric *vacuum* solutions, the traversable wormhole geometry cannot, so we need to generalise.

Let us work in Gaussian normal coordinate suited to the domain wall:

$$ds^2 = d\eta^2 + g_{ij}(x,\eta)dx^i dx^j\,, \tag{B.2}$$

with $x^i \in \{\tau, \theta, \phi\}$ the coordinates on the wall, and $\eta = 0$ the domain wall location. To determine the dynamics of the domain wall, we will need to calculate its extrinsic curvature, which has a simple form in GNC,

$$K_{ij} = \frac{1}{2}\partial_\eta g_{ij}\,, \tag{B.3}$$

and, in particular, we will need the $\theta\theta$ components,

$$K_{\theta\theta} = \frac{1}{2}\partial_\eta g_{\theta\theta} = \frac{1}{2}\partial_\eta r^2 = \frac{1}{2}\xi^\mu \partial_\mu r^2\,, \tag{B.4}$$

where $\xi^\mu$ is the unit normal to the wall. To find the components of $\xi^\mu$, and so determine $K_{\theta\theta}$, we use that the normal is orthogonal to the 4-velocity of the domain wall, $g_{\mu\nu}U^\mu \xi^\nu = 0$, where $U^\mu := (\dot{t}, \dot{r}, 0, 0)$, with the dot denoting differentiation with respect to the proper time $\tau$ of the

wall. Solving this orthogonality relation in combination with the normalisation conditions, $|U| = -1$ and $|\xi| = 1$, gives the components of the norm:

$$\xi^r = \pm\sqrt{f_2(r) + \dot{r}^2}, \quad \text{and} \quad \xi^t = \frac{\dot{r}}{\sqrt{f_1(r)f_2(r)}}. \tag{B.5}$$

The component $\xi^r$ is also sometimes denoted by $\beta$. The domain wall effective potential comes from the $\theta\theta$ component of the junction condition (in GNC)[15]

$$\Delta K^i_j = -8\pi G\left(S^i_j - \frac{1}{2}\delta^i_j \text{Tr}S\right), \tag{B.6}$$

which is

$$\Delta K_{\theta\theta} = -\kappa g_{\theta\theta} = -\kappa r^2, \tag{B.7}$$

where

$$\Delta K^i_j := \lim_{\epsilon \to 0}\left(K^i_j(\eta = \epsilon) - K^i_j(\eta = -\epsilon)\right). \tag{B.8}$$

Here we have separated from the global stress tensor the term that is localised to the domain wall, known as the surface stress tensor

$$T^{\mu\nu} = S^{\mu\nu}(x^i)\delta(\eta) + \text{regular terms}, \tag{B.9}$$

and we have assumed that the surface stress tensor is that of a perfect fluid with a vacuum energy equation of state,

$$S_{ij} = -\frac{\kappa}{4\pi G}g_{ij}. \tag{B.10}$$

When $f_1 = f_2$, $K_{\tau\tau}$ is not an independent constraint on the wall dynamics than $K_{\theta\theta}$; the latter is the proper time derivative of the former [51]. It is not a priori clear whether this is still true in our geometry where $f_1 \neq f_2$. In [51], the functional interdependence follows from the conservation of the full stress tensor $T^{\mu\nu}_{;\mu} = 0$. Said another way, there is a functional dependence between components of the junction conditions because of the divergencelessness of the Einstein tensor. The stress tensor is non-zero away from the domain wall in our set-up, but it is still conserved.

The end result is that the equation of motion of a vacuum energy domain wall in the $f_1$-$f_2$ metric (B.1) is identical to the standard domain equation of motion with effective potential, except with $f \mapsto f_2$.

## C  Domain wall with vacuum equation of state

Let us assume the domain wall is composed of vacuum energy so that its energy density $\kappa$ is $r$-independent. The condition (224) is inconsistent with non-zero tension domain walls reaching the asymptotic boundary. The physical interpretation is the following: for a constant tension domain wall to reach the boundary requires an infinite amount of energy, and nucleating the

---

[15]A one paragraph review of how the Israel junction conditions are derived: Einstein's equations are written in Gaussian normal coordinates. The terms that can be singular on the domain wall are picked out. Continuity of the metric across the wall ensures the extrinsic curvature has no delta function singularity there. The equation quoted here is the $ij$ component of the Einstein's equations - the $i\eta$ and $\eta\eta$ components are automatically satisfied if $K_{ij}$ is non-singular.

traversable wormhole only gives us a finite, order $\Delta M$ amount of energy; the wall cannot reach the boundary by energy conservation. Furthermore, since our analysis is only valid for $R > \bar{r}$, the maximum tension domain wall the condition (224) allows us to consider is bounded by

$$\kappa^2 \leq \frac{\epsilon^2 \mathcal{C}(\bar{r})}{\bar{r}^2}. \tag{C.1}$$

Physically, this upper bound on the domain wall tension is controlled by $\epsilon$ because of the difference between the true and false vacuum energies; the wormhole geometry is approximately super-extremal AdS-RN in the throat, and $\epsilon$ quantifies the distance from extremality.

We would like to know how many positive real roots the effective potential can have; roots are turn-around radii for domain walls. $r^4 V_{eff}(r)$ is a sixth-order polynomial so a priori there could be up to six roots. To reduce the set of possible domain wall trajectories and simplify our analysis, we use Descartes' sign rule: the maximum number of the positive real roots of a polynomial equals the number of sign changes between consecutive coefficients in the polynomial expansion. We expand $r^4 V_{eff}(r)$ in powers of $(r - \bar{r})$, as we are not interested in roots where $r < \bar{r}$. For arbitrary $\ell$, $\bar{r}$ and $\kappa$, we can show that there can be at most two sign changes - at most two roots in the effective potential. For certain ranges of $\kappa$, the sign rule can further restrict the maximum number of roots:

| Domain wall tension | Number of roots of $V_{eff}(r)$ |
| :---: | :---: |
| $\kappa^2 < \frac{4}{\ell^2}$ | 1 |
| $\frac{4}{\ell^2} < \kappa^2 < \frac{4}{\ell^2} + \frac{2}{5\bar{r}^2}$ | 0 or 2 |
| $\frac{4}{\ell^2} + \frac{2}{5\bar{r}^2} < \kappa^2$ | 2 |

We require (C.1), for which only the $\kappa^2 < \frac{4}{\ell^2}$ case is consistent. The potential for this range of domain wall tension has exactly one root, and $\lim_{r \to \infty} V_{eff}(r) = +\infty$, so this potential confines the domain wall to $r$ less than the root of $V_{eff}(r)$. The domain wall always collapses.

When is the domain wall confined to the throat region? In the throat region, $\epsilon \ll (r - \bar{r})/\bar{r} \ll 1$, the effective potential to quadratic order is

$$
\begin{aligned}
V_{eff}(r) = {}& \mathcal{C}(\bar{r})\left(\frac{r - \bar{r}}{\bar{r}}\right)^2 - \frac{\epsilon^4 \mathcal{C}(\bar{r})^2}{4\kappa^2 \bar{r}^2}\left(1 - 4\left(\frac{r - \bar{r}}{\bar{r}}\right) + 10\left(\frac{r - \bar{r}}{\bar{r}}\right)^2\right) \\
& - \frac{\kappa^2 \bar{r}^2}{4}\left(1 + 2\left(\frac{r - \bar{r}}{\bar{r}}\right) + \left(\frac{r - \bar{r}}{\bar{r}}\right)^2\right) + \mathcal{O}\left(\left(\frac{r - \bar{r}}{\bar{r}}\right)^3\right).
\end{aligned}
\tag{C.2}
$$

Whether the domain wall is trapped in the throat region depends on the scaling of its tension with $\epsilon$. If $\kappa \ll \epsilon$, then the effective potential is negative and without roots in the throat region - the domain wall escapes the throat, though it still cannot reach the asymptotic boundary. If $\kappa \sim \epsilon$, then the effective potential has a root at

$$\frac{r - \bar{r}}{\bar{r}} = \epsilon\left(\frac{\mathcal{C}(\bar{r})}{4}\left(\frac{\kappa \bar{r}}{\epsilon}\right)^{-2} + \frac{1}{\mathcal{C}(\bar{r})}\left(\frac{\kappa \bar{r}}{\epsilon}\right)^2\right)^{1/2} + \mathcal{O}(\epsilon^2), \tag{C.3}$$

which allows for a turnaround radius in the throat. Recall that we still need $(r - \bar{r})/\bar{r} \gg \epsilon$, which further restricts the tension to be far smaller than the maximum allowed in (C.1).

In summary, for any $\kappa^2 \ll \epsilon^2 \mathcal{C}(\bar{r})/\bar{r}^2$, there is an allowed vacuum energy domain wall containing a bubble of wormhole geometry which inevitably collapses towards $r \lesssim \bar{r}$ where the wormhole geometry is no longer approximately super-extremal AdS-RN and we need a new approach in our analysis. While we know what the relevant geometries are in this range of $r$, we need to use the domain wall trajectory tools of appendix B, and we have not determined whether the bubble of wormhole geometry will completely collapse or whether it will oscillate.

# D   Perfect fluid domain wall trajectories

We want to determine the thin domain wall dynamics when the wall is composed of a general perfect fluid; the literature has predominantly focused on the case where the perfect fluid is vacuum energy.

Our stress tensor has three terms: one each for outside, on, and inside the domain wall:

$$T^{\mu\nu}(x) = S^{\mu\nu}(x^i)\delta(\eta) + T^{\mu\nu}_{out}(x)\theta(\eta) + T^{\mu\nu}_{in}(x)\theta(-\eta). \tag{D.1}$$

The junction condition

$$\Delta K^i_j = -8\pi G\left(S^i_j - \frac{1}{2}\delta^i_j \mathrm{Tr}S\right), \tag{D.2}$$

is true for arbitrary $S^i_j$.

Stress tensor conservation gives us

$$0 = \nabla_\nu T^{i\nu} = \left(\nabla^{(3)}_j S^{ij} + 2K^i_j S^{j\eta} + Tr(K)S^{i\eta} + T^{i\eta}_{out} - T^{i\eta}_{in}\right)\delta(\eta)S^{i\eta}\delta'(\eta) + \text{regular terms}. \tag{D.3}$$

In order for this to be satisfied, it is required that require $S^{i\eta} = 0$ and

$$\nabla^{(3)}_j S^{ij} + T^{i\eta}_{out} - T^{i\eta}_{in} = 0. \tag{D.4}$$

This implies the conservation of the domain wall's surface stress tensor if the momentum flux across the wall is continuous

$$T^{i\eta}_{out} = T^{i\eta}_{in}. \tag{D.5}$$

When the stress tensor in and outside the wall is that of vacuum, so that it is proportional to the metric, then (D.5) is satisfied because $g^{i\eta} = 0$.

The surface stress tensor is given by:

$$4\pi G S^{\mu\nu} = \kappa(\tau)U^\mu U^\nu - \zeta(\tau)((g^{\mu\nu} - n^\mu n^\nu) + U^\mu U^\nu), \tag{D.6}$$

where $U^\mu$ is the 4-velocity of the domain wall, $n^\mu$ the unit normal, $\kappa$ the energy density,[16] and $\zeta$ the tension. Plugging this into the conservation equation $\nabla^{(3)}_j S^{ij} = 0$ gives

$$\dot{\kappa} = -2(\kappa - \zeta)\frac{\dot{r}}{r}. \tag{D.7}$$

We assume the equation of state

$$\zeta = \left(1 - \frac{\alpha}{2}\right)\kappa, \tag{D.8}$$

and solve the conservation equation (D.7) to get

$$\kappa(r) \propto r^{-\alpha}. \tag{D.9}$$

If the domain wall is composed of dust, then the surface tension is zero, so $\alpha = 2$. $\alpha = 3$ corresponds to radiation, for which $S^{\mu\nu}$ is traceless. A domain wall composed of vacuum energy has an energy density equal to its tension, so $\alpha = 0$ and its energy density is radius-independent.

We would like to check whether the equation of motion for the domain wall depends on its equation of state. If we plug the general perfect fluid surface stress tensor (D.6) into the $\theta\theta$ component of the junction condition (D.2) we get

$$\Delta K^\theta_\theta = -\kappa. \tag{D.10}$$

---

[16]Strictly speaking, $\kappa$ is only proportional to the energy density, see its definition in (B.10).

This junction condition does not depend on the domain wall tension $\zeta$ and it is true for any domain wall equation of state i.e. whether it is composed of dust, radiation, or vacuum energy. The domain wall dynamics follow from (D.10) without further assumptions about the composition of the wall.

# E  Non-static domain walls with radiation equation of state

Here we analyse the general dynamics of radiation-composed domain walls glueing together the traversable wormhole and the extremal AdS-RN geometries. For a radiation-composed domain wall, with energy density $\kappa = \sigma r^{-3}$, the domain wall effective potential as a power expansion in $r$ is

$$V_{eff}(r) = \left( \frac{1}{\ell^2} - \frac{G^2 \Delta M^2}{\sigma^2} \right) r^2 + 1 - \frac{2GM + G\Delta M}{r} + \frac{r_e^2}{r^2} - \frac{\sigma^2}{4r^4} \,. \tag{E.1}$$

Equation (224), the condition that needs to be satisfied to be glueing the wormhole interior to the black hole exterior, for $\alpha = 3$ becomes

$$R^3 \geq \frac{\sigma^2 \ell}{2\sigma_c} \,. \tag{E.2}$$

This rules out finite energy density domain walls collapsing to zero size. $\sigma_c$ is the critical value for $\sigma$ that determines the large $r$ behaviour of the potential:

$$\lim_{r \to \infty} V_{eff}(r) = \begin{cases} +\infty\,, & \text{if } \sigma > \sigma_c\,, \\ -\infty\,, & \text{if } \sigma < \sigma_c\,, \end{cases} \quad \text{with} \quad \sigma_c := G|\Delta M|\ell = \frac{\epsilon^2}{2}\mathcal{C}(\bar{r})\bar{r}\ell \,. \tag{E.3}$$

We can rewrite (E.1) in a form that will be convenient for approximations in the throat region, in terms of $\sigma_c$ and taking advantage of $M$ and $r_e$ being related for an extremal black hole,

$$V_{eff}(r) = \frac{1}{\ell^2} \left( \frac{r - \bar{r}}{r} \right)^2 (\ell^2 + r^2 + 2r\bar{r} + 3\bar{r}^2) - \left( \frac{\sigma_c}{\sigma} \frac{r}{\ell} - \frac{\sigma}{2r^2} \right)^2 \,. \tag{E.4}$$

The first term is the emblackening factor of an extremal AdS RN black hole.

## E.1  Sub-critical energy density

Let us first consider radiation domain walls with sub-critical energy densities, $\sigma < \sigma_c$, whose distinguishing feature is that $V_{eff}(r = \infty) = -\infty$. The condition (E.2) is satisfied everywhere in the relevant domain, $(r - \bar{r})/\bar{r} \gg \epsilon$, as the right-hand side is subleading in $\epsilon$. In the effective potential (E.4), the $\sigma/2r^2$ term can be neglected at leading order in $\epsilon$ for the domain $(r - \bar{r})/\bar{r} \gg \epsilon$. This simplified effective potential is proportional to a quartic polynomial in $r$, and by calculating the polynomial's Sturm sequence we can show that the potential has at most two roots in the domain $\bar{r} < r$, and that it can only have two roots when $\sigma < \sigma_c$. These two roots can be found exactly. Their perturbative expansions as $\sigma$ approaches criticality are

$$r_1 = \bar{r} \left( 1 + \frac{\bar{r}}{\ell} \left( 2\frac{\bar{r}}{\ell} + \sqrt{1 + 4\frac{\bar{r}^2}{\ell^2}} \right) \right) + \mathcal{O}\left( \sqrt{\frac{\sigma_c}{\sigma} - 1} \right) \,, \tag{E.5}$$

and

$$r_2 = \frac{\ell}{\sqrt{2\left( \frac{\sigma_c}{\sigma} - 1 \right)}} - \bar{r} \left( 1 + \frac{2\bar{r}^2}{\ell^2} \right) + \mathcal{O}\left( \sqrt{\frac{\sigma_c}{\sigma} - 1} \right) \,. \tag{E.6}$$

The effective potential is negative above $r_2$ and below $r_1$, and positive between. In the Lorentzian section, $r_2$ is the closest-approach radius for a domain wall going to or from the asymptotic boundary. The turnaround radius becomes arbitrarily large as $\sigma$ approaches its critical value (from below), so sub-critical $\sigma$ guarantees that the domain wall stays out near the asymptotic boundary, away from the wormhole mouth. Solving the domain wall equation of motion $\dot{R}^2 + V_{eff}(R) = 0$ in the large $R$ regime gives the Lorentzian trajectory:

$$R_L(\tau) \approx \frac{\ell}{\sqrt{2\left(\frac{\sigma_c}{\sigma} - 1\right)}} \cosh\left(\frac{\sqrt{2\left(\frac{\sigma_c}{\sigma} - 1\right)}}{\ell}\tau\right),$$ (E.7)

where $\tau$ is the (Lorentzian) proper time of the wall. This was derived by solving the equation of motion after dropping terms that are $O(R^{-1})$ in $V_{eff}(R)$. The domain wall accelerates out to the asymptotic boundary.

When the domain wall is at the turnaround radius $r_2$, we are at a point of time-reflection symmetry and we can Wick rotate to Euclidean signature. The trajectory of the domain wall in Euclidean signature is a gravitational instanton solution whose action determines the tunnelling rate. The effective potential that determines the trajectory of the domain wall in Euclidean signature is minus the effective potential of the domain wall in Lorentzian signature [86]. A domain wall that is initially at rest at $R = r_2$ will accelerate towards increasing $r$ in Lorentzian signature and decreasing $r$ in Euclidean signature. In Euclidean signature, the radial position of our domain wall will decrease monotonically from $R = r_2$ and oscillate within the interval $[r_1, r_2]$. For previous work on oscillating instantons, see [20, 87, 88].

The periodicity of the domain wall with respect to the asymptotic Euclidean time is

$$\int d\tau_o = 2 \int_{r_1}^{r_2} dR \frac{d\tau}{dR} \frac{d\tau_o}{d\tau}.$$ (E.8)

The two derivatives in this equation are known: the time dilation factor between the asymptotic time and domain wall proper time is given by

$$\begin{aligned}
\frac{d\tau_o}{d\tau} &= f_o^{-1}(R)\sqrt{f_o(R) - \dot{R}^2} \\
&= f_o^{-1}(R)\sqrt{f_o(R) - V_{eff}(R)},
\end{aligned}$$ (E.9)

with $f_o$ the emblackening factor of an extremal AdS-RN black hole, and the second line following from the domain wall equation of motion. The domain wall periodicity is logarithmically divergent as we approach the critical energy density:[17]

$$\int d\tau_o = \ell \log\left(\frac{\sigma_c}{\sigma_c - \sigma}\right) + \mathcal{O}(1).$$ (E.10)

## E.2 Throat region

To satisfy (E.2) when the domain wall is in the throat region, i.e. when $R \sim \bar{r}$, requires[18]

$$\frac{\sigma}{\sigma_c} \leq \begin{cases} \sqrt{\frac{2}{3}}\frac{1}{\epsilon}, & \text{for } \bar{r} \gg \ell, \\ \frac{\bar{r}}{\ell}\sqrt{\frac{2}{\epsilon}}, & \text{for } \bar{r} \ll \ell, \end{cases}$$ (E.11)

---

[17] One method for performing the integral over $R$ is to first transform $2R \mapsto (r_1 + r_2) + (r_2 - r_1)\sin(\theta)$. This is the standard transformation for integrals of the form $\int_{x_1}^{x_2} dx((x - x_1)(x_2 - x))^{-1/2} f(x)$. The resulting denominator is the sum of two terms, one of which can be dropped because it is subleading $\sigma_c - \sigma$ for all $\theta$, though it is not obvious to show around $\theta = -\pi/2$. Then one integrates over $\theta$ and expands in $\sigma_c - \sigma$.

[18] We need to take $\bar{r}/\ell$ to be either perturbatively small or large in order to write $\sigma_c$ as a simple function of $\bar{r}$.

which is not a strong constraint on $\sigma$. In the throat region, the $\alpha = 3$ effective potential to quadratic order in $(r - \bar{r})/\bar{r} \ll 1$ is[19]

$$V_{eff}(r) = \mathcal{C}(\bar{r}) \left( \frac{r - \bar{r}}{\bar{r}} \right)^2 - \frac{\sigma_c^2}{\sigma^2} \frac{r^2}{\ell^2} \, . \tag{E.12}$$

For this potential to have a root requires

$$\frac{\sigma}{\sigma_c} \gg \frac{\bar{r}}{\ell \sqrt{\mathcal{C}(\bar{r})}} \, , \tag{E.13}$$

and this root is at

$$\tilde{r}_t = \frac{\bar{r}}{1 - \frac{\sigma_c}{\sigma} \frac{\bar{r}}{\ell \sqrt{\mathcal{C}(\bar{r})}}} \, . \tag{E.14}$$

There is no value of $\sigma/\sigma_c$ for which the domain wall potential has both a root in the throat region and the large $r$ region; (E.13) is not compatible with $\sigma < \sigma_c$. If $\sigma < \sigma_c$ then the effective potential is negative throughout the throat region, and there must be a second root at or outside the mouth of the wormhole's throat, *i.e.* in the region $(r - \bar{r}) \gtrsim \bar{r}$.

# F  Perturbations around static domain wall solutions

To really confirm whether the static domain wall gives the correct decay rate, one needs to verify whether this convenient, static instanton is actually dominant. Note that only a discrete set of solutions is allowed, due to the periodicity in imaginary time. In addition to our static solution, one can also consider solutions that oscillate around the minimum of the Euclidean potential. While these are difficult to compute, a useful test is whether our static solution has the correct number of negative modes- namely, only one. Having more negative modes is an indication that we have not found the dominant instanton because we can flow down along these negative directions to an instanton with lower action.

To do this, we just need to expand our equations around the static location $\hat{r}$. For this, it is most convenient to write the action in terms of an integral over Euclidean time,

$$S = 4\pi\sigma \int \frac{d\tau}{r} = 4\pi\sigma \int \frac{dt}{r} \sqrt{f + f^{-1} \left( \frac{dr}{dt} \right)^2} \, . \tag{F.1}$$

Note the different sign inside the square root because we are in Euclidean signature.

We are interested in small perturbations around the static solution, so the time derivative will be small, and we can expand to get

$$S = 4\pi\sigma \int \frac{dt}{r} \left[ \sqrt{f} + \frac{1}{2} f^{-3/2} \left( \frac{dr}{dt} \right)^2 \right] \, . \tag{F.2}$$

Now we expand for $r$ near $\hat{r}$. Define $r = \hat{r} + \delta r$. Note that the condition for $\hat{r}$ guarantees that the function $\sqrt{f}/r$ has zero derivative at $\hat{r}$, so to second order in the perturbations,

$$S = 4\pi\sigma \int dt \left[ \frac{\sqrt{\hat{f}}}{\hat{r}} + \frac{1}{2} \frac{d^2}{dr^2} \left( \frac{\sqrt{f}}{r} \right) \bigg|_{\hat{r}} (\delta r)^2 + \frac{1}{2\hat{r}} \hat{f}^{-3/2} \left( \frac{d(\delta r)}{dt} \right)^2 \right] \, . \tag{F.3}$$

---

[19]The $\sigma_c r^{-1}$ and $\sigma^2 r^{-4}$ terms that we have dropped are subleading in $\epsilon$ with respect to the $\sigma_c^2 r^2 / \sigma^2 l^2$ term as long as $\sigma/\sigma_c \ll \epsilon^{-1}$, i. e. as long as we are no close to saturating the upper bound allowed for large black holes in (E.11). In *that* edge case, all $\sigma$ and $\sigma_c$-dependent terms are subleading in $\epsilon$ with respect to the first term in (E.12), and there are no roots.

The first term is the action for the static wall. The last term is a positive kinetic term, but the middle term is negative. We can analyze this by Fourier transforming in time. Recalling the periodicity, the allowed frequencies are

$$\omega = \frac{2\pi n}{\beta}, \tag{F.4}$$

for integer $n$. There is clearly one negative mode, coming from $\omega = 0$. In order for the static instanton to dominate, the other modes should be positive. This requires

$$\frac{\hat{f}^{-3/2}}{\hat{r}}\left(\frac{2\pi n}{\beta}\right)^2 > -\frac{d^2}{dr^2}\left(\frac{\sqrt{f}}{r}\right)\Big|_{\hat{r}}, \tag{F.5}$$

for all nonzero $n$.

For Schwarzschild black holes, this can be checked explicitly and it is satisfied, indicating that the static instanton dominates. This extends to near-Schwarzschild black holes since the inequality is not saturated for Schwarzschild. However, for near-extremal black holes, it appears that $\beta \to \infty$ while the other quantities in the above equation remain finite. In particular, for an extremal black hole $\hat{r} = 2r_e$ and

$$\frac{d^2}{dr^2}\left(\frac{\sqrt{f}}{r}\right)\Big|_{\hat{r}} = -\frac{1}{\hat{r}^3}. \tag{F.6}$$

Therefore, it appears that for near-extremal black holes, the static instanton is not dominant. This is puzzling because it appears to give the 'correct' answer. Perhaps the additional negative modes develop when the charge is large enough that the RN black hole begins to have a positive specific heat. Schwarzschild black holes have negative specific heat, but above some critical charge, the specific heat becomes positive. In this regime, perhaps they are in a sense no longer unstable to emitting neutral particles. In addition, it is known that the number of negative modes of the background solution changes at this threshold.

Incidentally, generalizing to RNAdS black holes we get,

$$\hat{r} = 2\frac{r_e^2}{\bar{r}}, \tag{F.7}$$

and (F.5) becomes

$$\left(\frac{2\pi n}{\beta}\right)^2 > \frac{\left(1 + 2\frac{\bar{r}^2}{\ell^2}\right)\left(1 + 6\frac{\bar{r}^2}{\ell^2}\right)^4}{64\bar{r}^3\left(1 + 3\frac{\bar{r}^2}{\ell^2}\right)^5}. \tag{F.8}$$

This expression is consistent with (F.5). Note from (F.7) that when the black hole becomes large compared to the AdS radius, the static solution disappears.

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
