# Peer review of "Tunnelling to Holographic Traversable Wormholes"

_SciPost Physics, doi:SciPost Phys. 16, 066 (2024)_

## Round 1 · Referee Report · Anonymous (Referee 1) · 2023-10-31

Strengths

  1. This paper computes an interesting non-perturbative effect describing tunneling between three different solutions in AdS.
  2. The computations are explained well with an in-depth discussion of the different ensembles/boundary conditions.
  3. Where relevant this work compares and contrasts with other results.

Weaknesses

  1. I am somewhat skeptical that these results are relevant for the motivation mentioned in the introduction, i.e. testing gravity using holographic simulations (see the report).
  2. Some effects that might be relevant (e.g. Schwinger pair production) are not discussed.

Report

This paper studies the tunneling rate between three different solutions in AdS$_4$ when coupling the two boundary CFTs: empty AdS, two (near-)extremal magnetic black holes and a traversable wormhole. This is an interesting computation that is relevant to understand non-perturbative (quantum) gravity effects in holography. The results are presented clearly and as far as I can tell and have checked the computations are correct. I see a two aspects that can be improved.

First, I think the results are of interest on its own, but the authors mention in the introduction that they might be relevant for "testing gravity predictions" by simulating holographic CFTs on a quantum computer. To see these (exponentially) suppressed effects we have to work in a regime where there is no semi-classical bulk dual. Can we even sharply distinguish between the different bulk solutions in this case?

Second, the authors study an extremal magnetic black hole solution with massless charged Dirac fermions. They view these solutions as ground states, but I expect these solutions to be unstable under Schwinger pair production. A discussion how this influences their results (or why it does not) would be useful.

Requested changes

  1. Clarify precisely what the authors have in mind when they mention they want to test gravity outside of the semi-classical regime using holographic simulations. Is there a sharp prediction they want to test? Can we even define these different (semi-)classical solutions?

  2. Explain why the extremal black holes can be viewed as ground states. If not, why is Schwinger pair production not a relevant decay channel.

  3. Add a reference to https://arxiv.org/abs/2004.06084. This paper showed that near-extremal magnetic black holes are highly unstable. This seems relevant for the results of this paper for near-extremal black holes such as in Sec. 5.1.

---

## Round 2 · Referee Report · Anonymous (Referee 1) · 2024-2-13

Report

The authors addressed all feedback I had. I am therefore happy to recommend this paper for publication.

---

## Round 2 · Referee Report · Anonymous (Referee 2) · 2024-2-15

Weaknesses

There is perhaps too much emphasis on the possibility of test experimentally the result and simulations, but it is confined in the discussion session.

Report

The paper computes the tunneling rate from a pair of disconnected extremal black holes to a traversable wormhole. This is a non-perturbative effects of quantum gravity involving a change of topology. This is an interesting computation for understanding quantum gravity effects in holography. This timely paper is clear and well-written and, in my opinion, meets the criteria for publication in SciPost.

---

## Round 2 · Author Response

We thank the referee for their insightful questions and comments. Below we address each of the comments in turn.

  1. Clarify precisely what the authors have in mind when they mention they want to test gravity outside of the semi-classical regime using holographic simulations. Is there a sharp prediction they want to test? Can we even define these different (semi-)classical solutions?

Let us first clarify what we mean by semi-classical. The solutions, like the traversable wormhole, are well-defined semiclassical solutions (in the sense of being solutions to Einstein’s equations using ⟨Tμν ⟩). What is outside of the semiclassical regime is the instanton tunnelling rate, in the sense that it is non-perturbative in GN ℏ, and that rate is the prediction that we would in principle want to test. For fixed temperature, the canonical ensemble, the rate is given by (1.4), while for fixed energy, the microcanonical ensemble, it is given by (1.3). Within our regime of validity, these tunnelling rates are small, exponentially suppressed, but non-zero.

We have added a sentence towards the end of the third paragraph of the introduction for clarification.

  1. Explain why the extremal black holes can be viewed as ground states. If not, why is Schwinger pair production not a relevant decay channel.

The semi-classical solution which is a pair of extremal black holes is not the true ground state outside of the bottom right of the region of parameter space shown in our Fig 1. We explicitly compare the energies of the different solutions in Sec 3 and find the regions of parameter space where either the traversable wormhole or empty AdS solutions have lower energy. It is true that AdS-RN black holes can develop instabilities as they approach extremality, zero temperature, as investigated for example in [1–7]. We are investigating another kind of instability; in our paper, we are calculating the fixed-energy and temperature rates for an instanton-mediated decay channel. As the referee points out, an important question is whether this decay channel is the dominant one. We considered other decay channels, such as in section 3.4 where we consider decay via fragmentation. With regards to the specific concern of the decay of our black hole via Schwinger pair production, presumably of magnetic monopoles, we assume that the mass of such monopoles is sufficiently large that this decay channel is irrelevant. The decay rate is exponentially suppressed by the mass of the monopole, and heavy monopoles are confined by the effective radial potential.

We have added a new subsection, section 3.5, going into more detail on other decay channels.

  1. Add a reference to https://arxiv.org/abs/2004.06084. This paper showed that near-extremal magnetic black holes are highly unstable. This seems relevant for the results of this paper for near-extremal black holes such as in Sec. 5.1.

The suggested reference discusses an enhancement of Hawking radiation of near-extremal magnetic black holes. In sec 5.1, we are working in the canonical ensemble, so the macrostate is in thermodynamic equilibrium. For the black hole phase, this means that emitted Hawking radiation, even if enhanced, is in equilibrium with incoming thermal radiation, so we do not believe that Maldacena’s enhancement result is of direct relevance to our work. The underlying reason is that in our setup the geometries we consider are asymptotically AdS, while Maldacena’s results are valid in asymptotically flat space.

We have added the reference and this comment to the new section 3.5.

References

[1] S. S. Gubser, “Breaking an Abelian gauge symmetry near a black hole horizon,” Phys. Rev. D 78 (2008) 065034, arXiv:0801.2977 [hep-th]. [2] S. A. Hartnoll, C. P. Herzog, and G. T. Horowitz, “Building a Holographic Superconductor,” Phys. Rev. Lett. 101 (2008) 031601, arXiv:0803.3295 [hep-th]. [3] S. A. Hartnoll, C. P. Herzog, and G. T. Horowitz, “Holographic Superconductors,” JHEP 12 (2008) 015, arXiv:0810.1563 [hep-th]. [4] S.-S. Lee, “A Non-Fermi Liquid from a Charged Black Hole: A Critical Fermi Ball,” Phys. Rev. D 79 (2009) 086006, arXiv:0809.3402 [hep-th]. [5] H. Liu, J. McGreevy, and D. Vegh, “Non-Fermi liquids from holography,” Phys. Rev. D 83 (2011) 065029, arXiv:0903.2477 [hep-th]. [6] M. Cubrovic, J. Zaanen, and K. Schalm, “String Theory, Quantum Phase Transitions and the Emergent Fermi-Liquid,” Science 325 (2009) 439–444, arXiv:0904.1993 [hep-th]. [7] T. Faulkner, H. Liu, J. McGreevy, and D. Vegh, “Emergent quantum criticality, Fermi surfaces, and AdS(2),” Phys. Rev. D 83 (2011) 125002, arXiv:0907.2694 [hep-th].

---

## Round 2 · List of Changes

Added a sentence in the third paragraph of the introduction to clarify the non-perturbative effect of gravity we probe: "In other words, we compute the non-perturbative, yet dominant decay rate between semi-classical solutions with different topologies."

Added a new section 3.5 on possible instabilities of the black hole phase such as Schwinger pair production and the enhancement of Hawking radiation of https://arxiv.org/abs/2004.06084.

---

## Editorial Decision

published